# Characterization of *in situ* cosmogenic [14]CO production, retention and loss in firn and shallow ice at Summit, Greenland

Benjamin Hmiel[1,a,*], Vasilii V. Petrenko[1,*], Christo Buizert[2], Andrew M. Smith[3], Michael N. Dyonisius[1,b], Philip Place[1,c], Bin Yang[3], Quan Hua[3], Ross Beaudette[4], Jeffrey P. Severinghaus[4], Christina Harth[4], Ray F. Weiss[4], Lindsey Davidge[1,d], Melisa Diaz[1,e], Matthew Pacicco[1,f], James A. Menking[2,g], Michael Kalk[2], Xavier Faïn[5], Alden Adolph[6,h], Isaac Vimont[7,i], Lee T. Murray[1]

[1]Department of Earth and Environmental Sciences, University of Rochester, Rochester, NY, USA
[2]College of Earth, Ocean and Atmospheric Sciences, Oregon State University, Corvallis, OR, USA
[3]Centre for Accelerator Science, Australian Nuclear Science and Technology Organization, Lucas Heights, NSW, Australia
[4]Scripps Institution of Oceanography, University of California, San Diego, La Jolla, CA, USA
[5]Univ. Grenoble Alpes, CNRS, INRAE, IRD, Grenoble INP, IGE, Grenoble, France
[6]Thayer School of Engineering, Dartmouth College, Hanover, NH, USA
[7]Institute of Arctic and Alpine Research, University of Colorado Boulder, Boulder, CO, USA
[a]present address: Air Pollution Control Division, Colorado Department of Public Health and the Environment, Glendale, CO, USA
[b]present address: Physics of Ice, Climate and Earth, Niels Bohr Institute, University of Copenhagen, Copenhagen, Denmark
[c]present address: University Instrumentation Center, University of New Hampshire, Durham, NH, USA
[d]present address: Earth and Space Sciences, University of Washington, Seattle, WA, USA
[e]present address: Byrd Polar and Climate Research Center, The Ohio State University, Columbus, OH, USA
[f]present address: Department of Chemistry and Physics, Monmouth University, West Long Branch, NJ, USA
[g]present address: Australian Antarctic Program Partnership, Institute for Marine and Antarctic Studies, University of Tasmania, Hobart, TAS, Australia
[h]present address: Department of Physics, St. Olaf College, Northfield, MN, USA
[i]present address: National Oceanic and Atmospheric Administration, Global Monitoring Laboratory, Boulder, CO, USA
*These authors contributed equally to the study

*Correspondence to:* Vasilii V. Petrenko (vasilii.petrenko@rochester.edu)

**Abstract**

Measurements of carbon-14-containing carbon monoxide ([14]CO) in glacial ice are useful for studies of the past oxidative capacity of the atmosphere as well as for reconstructing the past cosmic ray flux. [14]CO abundance in glacial ice represents the combination of trapped atmospheric [14]CO and *in situ* cosmogenic [14]CO. The systematics of *in situ* cosmogenic [14]CO production and retention in ice are not fully quantified, posing an obstacle to interpretation of ice core [14]CO measurements. Here we provide the first comprehensive characterization of [14]CO at an ice accumulation site (Summit, Greenland), including measurements in the ice grains of the firn matrix, firn air and bubbly ice below the firn zone. The results are interpreted with the aid of a firn gas transport model into which we implemented *in situ* cosmogenic [14]C. We find that almost all (≈99.5%) of *in situ* [14]CO that is produced in the ice grains in firn is very rapidly (in <1 year) lost to the open porosity and from there mostly vented to the atmosphere. The time scale of this rapid loss is consistent with what is expected from gas diffusion through ice. The small fraction of *in situ* [14]CO that initially stays in the ice grains continues to slowly leak out to the open porosity at a rate of ≈0.6% per year. Below the firn zone we observe an increase in [14]CO content with depth that is due to *in situ* [14]CO production by deep-penetrating

muons, confirming recent estimates of $^{14}$CO production rates in ice via the muon mechanisms and allowing for
narrowing constraints on these production rates.

**1 Introduction to $^{14}$C in glacial ice and firn and motivation for studies of $^{14}$CO in ice**

Measurements of carbon-14 ($^{14}$C) in ice cores have been explored in a number of applications, including the
determination of glacial ablation rates (e.g., Lal et al., 1990) and ice core dating (e.g., van de Wal et al., 2007; Wilson
and Donahue, 1992). More recently, they have been used for characterization of the fossil fraction of the
paleoatmospheric methane budget (Petrenko et al., 2017; Dyonisius et al., 2020; Hmiel et al., 2020). $^{14}$C is included
in glacial ice via two separate mechanisms: trapping of carbon-containing atmospheric gases (mainly carbon dioxide
($CO_2$), carbon monoxide (CO) and methane ($CH_4$)) into air bubbles in ice and *in situ* production directly from $^{16}$O
within the ice crystal lattice via neutron-induced spallation (e.g., Lal et al., 1987), negative muon capture (van der
Kemp et al., 2002) and interactions with fast muons (Petrenko et al., 2016). The *in situ* produced $^{14}$C in the gas phase
mainly forms $^{14}$CO$_2$ and $^{14}$CO, with a smaller fraction forming $^{14}$CH$_4$; other simple organics may account for up to
25% of total *in situ* $^{14}$C (Dyonisius et al., 2023; Hoffman, 2016; van der Kemp et al., 2002; Fang et al., 2021).

Applications of ice core $^{14}$C measurements depend on the ability to separately characterize the contributions
from the *in situ* and trapped atmospheric components. However, in glacial ice, these two $^{14}$C components exist in a
combined form and cannot be separated analytically. Therefore, for applications where the trapped atmospheric $^{14}$C
component is of interest, the *in situ* component must be independently constrained and corrected for, and vice versa.
The *in situ* $^{14}$C content in ice at a snow accumulation site is influenced by both production of $^{14}$C and its retention in
the firn grains. *In situ* production rates are highest near the surface (Fig. 1), but $^{14}$C produced in the firn layer can be
lost to the atmosphere via the interconnected porosity (e.g., Petrenko et al., 2013b; van der Kemp et al., 2000). While
the balance of evidence now indicates that most *in situ* $^{14}$C produced in the firn is not retained (Petrenko et al., 2013b
and references therein), there has been no work to date that has been able to fully characterize the production and
movement of *in situ* $^{14}$C throughout the entire firn column.

Of the trace gas species known to be affected by *in situ* $^{14}$C in ice ($CO_2$, CO and $CH_4$), the ratio of *in situ* to
trapped atmospheric $^{14}$C is highest for $^{14}$CO, making this the best target for studies of *in situ* $^{14}$C processes. One of the
main aims of this study is therefore to use new measurements of $^{14}$CO in the firn matrix (ice grains) and open porosity
(firn air) to provide a detailed quantitative characterization of *in situ* $^{14}$C retention in and leakage from the ice grains
in firn. In ice below the firn zone, $^{14}$CO is also of interest for paleoenvironmental applications. The trapped
atmospheric component of $^{14}$CO can provide information about the past atmospheric oxidative capacity because
atmospheric $^{14}$CO serves as an integrating proxy for hydroxyl radical abundance (e.g., Jöckel and Brenninkmeijer,
2002; Petrenko et al., 2021).  The *in situ* component of $^{14}$CO in ice is also of interest for studies of the past galactic
cosmic ray (GCR) flux, because at sites with low snow accumulation rates the *in situ* component is much larger than
the trapped atmospheric component and has the potential to record GCR changes without the confounding influences
of solar, geomagnetic, climate and carbon cycle variations (Petrenko et al., 2024).

All *in situ* cosmogenic $^{14}$C (including $^{14}$CO) that is produced below the firn zone is retained in the ice, and at
these larger depths the $^{14}$C production occurs via the muon mechanisms only (Fig. 1). The interpretation of ice core

$^{14}$CO measurements for the above applications therefore requires a good understanding of the muogenic $^{14}$CO production rates. Dyonisius et al. (2023) and Petrenko et al. (2016) made measurements of $^{14}$CO in relatively shallow (0 – 72 m) ice obtained from the ablation zone of Taylor Glacier, Antarctica. The relatively old age (>50 ka) of this ice ensured that $^{14}$CO in the samples originated exclusively from *in situ* cosmogenic production during ice transport
within the glacier, allowing for constraints on muogenic $^{14}$CO production rates. However, relatively large uncertainties in ice flow trajectories within the glacier translated into relatively large uncertainties in $^{14}$CO production rates (up to 30% for the fast muon mechanism). Dyonisius et al. (2023) measured $^{14}$CO$_2$ and $^{14}$CH$_4$ in addition to $^{14}$CO and found that the overall *in situ* cosmogenic $^{14}$C production rate for these species in ice by muons is 4 – 5 times smaller than the total $^{14}$C production rate predicted from prior laboratory and field studies in quartz (Heisinger et al., 2002a;
Heisinger et al., 2002b; Lupker et al., 2015). The second main aim of this study is therefore to test the Dyonisius et al. (2023) muogenic $^{14}$CO production rates in ice at a site with very different characteristics and to place tighter constraints on these production rates.

**2 Sample Collection, Analyses and Calculation of $^{14}$CO concentration**

**2.1 Overview of Samples**

The study site (72.66˚ N, 38.58˚ W, 3214 m above sea level) was located ≈10 km to the NW of Summit station in central Greenland. Four different types of samples were collected for use in this study: (1) firn air samples, (2) firn matrix samples containing only negligible amounts of trapped air, (3) combined firn matrix and bubbly ice samples within the lock-in zone (LIZ, the depth range where most of the bubble trapping process happens (e.g., Buizert, 2013); ≈68-80m at Summit) and (4) samples of bubbly ice below the LIZ to ≈135 m depth.

**2.2 Firn air sample collection and analyses**

     Firn air samples were collected in May 2013 from a borehole drilled with a 3" electromechanical ice core drill at depth levels between 0 and 80m (Table S1). Firn air was sampled using the US Firn Air Sampling Device (FASD; Battle et al., 1996) following established techniques (Buizert et al., 2012). For the purposes of $^{14}$CO
measurements, air samples and accompanying procedural blanks were collected into 35L electropolished stainless steel canisters. For the $^{14}$CO procedural blanks, clean ambient air from a cylinder was run into the FASD through a tube containing Sofnocat 423 reagent (Molecular Products) to remove all CO, including $^{14}$CO. CO dry air mole fraction ($x_{CO}$) in the $^{14}$CO samples was measured at the University of Rochester (UR) by gas chromatography (GC) with hot mercuric oxide reduction and photometric absorption (Peak Performer 1 from Peak Labs) and δ$^{13}$C-CO was measured
at the Institute of Arctic and Alpine Research (INSTAAR) (Vimont et al., 2017; Table S1). Additional firn air samples were also collected into 2.5L glass flasks and 34L stainless steel canisters for a range of supporting measurements (not shown) that allowed to characterize firn gas transport at this site. Further details on firn air samples and associated analyses were presented in the Supplementary Information of Hmiel et al. (2020).

### 2.3 Firn matrix and LIZ sample collection

Samples for investigating the *in situ* cosmogenic $^{14}CO$ content in the firn matrix (ice grains) shallower than 60 m (Table S2) were collected in May-June 2014 and 2015 using techniques similar to prior large-volume firn and ice core $^{14}CO$ measurements (Petrenko et al., 2016; Petrenko et al., 2013b). Firn for the surface sample was cut in a snow pit with clean handsaws, while for deeper samples it was recovered from multiple adjacent boreholes using the 24 cm diameter Blue Ice Drill (Kuhl et al., 2014). The firn was loaded into a large 670L ice melter (Petrenko et al.,

2008; Petrenko et al., 2013b), the melter was evacuated to ≈3 mbar and then ≈130 mbar of ultrapure $N_2$ was added as a flush gas. The evacuate-flush sequence was then repeated, followed by a final evacuation to 1 – 2 mbar and a subsequent injection of a standard gas into the melter. The standard gas is needed as a carrier because the shallow firn matrix contains very little trapped air. The standard gas was passed through Sofnocat to remove all CO (including $^{14}CO$) for samples collected in 2015 (36 m and 53 m samples), but not for samples collected in 2014 (surface to 20 m;

Table S3). The firn was then melted, the gases were equilibrated between the headspace and the water by recirculating the air through the water via a bubbler manifold at the melter bottom for 30 min, and the headspace air was then transferred to 35 L electropolished stainless steel canisters using diaphragm pumps.

        Samples from the LIZ (Table S2) were obtained in May – June 2015 from two adjacent boreholes using the Blue Ice Drill. These samples contain more trapped air than shallower firn but less than bubbly ice below the firn

zone. While the procedure for the melt-extraction of gases for LIZ samples was almost identical to the procedure for firn matrix samples, the amount of standard gas added to the ice melter (via Sofnocat to remove CO and $^{14}CO$) for each melt-extraction was adjusted so that the effective total air content for these samples approximately matched the effective total air content from the firn matrix and bubbly ice samples (Table S2). Additionally, the standard gas used for the LIZ samples contained an artificially high mole fraction of Ne (≈1800 µmol mol$^{-1}$) to help constrain the fraction

of the collected sample air originating from the standard gas.

        Several procedural blank tests ("water blanks" in Tables S2 – S4) were collected in both 2014 and 2015 seasons as follows. Following a sample melt-extraction, the meltwater was purged with ultrapure air at ≈2 L STP / min flow via Sofnocat and the bubbler manifold for ≈75 min to remove any remaining dissolved $^{14}CO$. Next, a standard gas was introduced to the melter headspace, approximately matching the volume of air obtained in real melt-

extractions. For the 2014 field campaign, the same standard gas was used for the water blanks as for the firn matrix samples, without flowing the gas through Sofnocat in both cases (Table S3). This standard gas also contained ≈100 µmol mol$^{-1}$ of Kr and ≈50 µmol mol$^{-1}$ of Xe. For the 2015 campaign, two different standard gases were used for the water blanks, both of which were passed through Sofnocat when introduced to the melter (same as for all 2015 firn matrix and LIZ samples). The standard gas used for 2015 Water Blank 1 was the same standard gas as was used for

the 2015 firn matrix samples and also contained ≈100 µmol mol$^{-1}$ of Kr and ≈50 µmol mol$^{-1}$ of Xe. The artificially high Kr and Xe allow for accurate characterization of gas partitioning between the water and headspace in the ice melter (Petrenko et al., 2016; Petrenko et al., 2013b). The standard gas used for 2015 Water Blank 2 was the same standard gas as was used for the 2015 LIZ samples.

**2.4 Bubbly ice sample collection and analyses of air from firn matrix, LIZ and bubbly ice samples**

Samples of bubbly ice (ice below the LIZ) were collected in May – June 2015 from 2 – 3 adjacent boreholes using the Blue Ice Drill (Table S2). Sample handling and melt-extraction of air for bubbly ice samples was very similar to the procedures for firn matrix samples and LIZ samples described above, except that no standard gas was added to the melter as a carrier.

Air samples obtained from melt-extractions of firn matrix, LIZ and bubbly ice and accompanying procedural blanks were measured for $\delta^{13}$C-CO at INSTAAR (Table S3) (Vimont et al., 2017), as well as for mole fractions of $SF_6$, CFC-11 and CFC-12 at the Scripps Institution of Oceanography (SIO; not shown) to confirm the absence of significant contamination by ambient air. $x_{CO}$ in 2014 samples was measured by gas chromatography (GC) with hot mercuric oxide reduction and photometric absorption (Peak Performer 1 from Peak Labs) and in 2015 samples by

cavity ring-down spectroscopy (Picarro G2401) at UR (Table S3). $\delta$Xe/Kr, $\delta$Xe/Ar, $\delta$Kr/Ar and $\delta$Ar/$N_2$ were measured at SIO (Bereiter et al., 2018; not shown) in all samples and blanks that were collected with the use of standard gases that contained high Kr and Xe. $\delta$Ne/$N_2$ was also measured at SIO in 2015 LIZ samples and 2015 Water Blank 2, which were all collected with the use of the high-Ne standard gas (not shown).

**2.5 Sample processing and measurement for $^{14}$CO**

      Methods used for processing and measurements of $^{14}$CO in ambient air (Petrenko et al., 2021), firn air (Hmiel et al., 2020) and ice core air samples (Dyonisius et al., 2020; Hmiel et al., 2020) have been previously described; here we provide a brief summary and details specific to the samples presented in this study. In a first step, air in the 35L canisters containing samples and field procedural blanks is diluted at UR to reduce the $^{14}$C activity of CO, bringing

the $^{14}$C activity into the range of commonly used $^{14}$C measurement standards (0 – 135 percent Modern Carbon (pMC)). This is accomplished via the addition of a standard gas ("dilution gas" in Tables S1 and S3) that has high $x_{CO}$ (10.29 $\pm$ 0.13 µmol mol$^{-1}$) and low (< 1 pMC) $^{14}$C activity. The dilution also increases the CO carbon mass (to $\approx$33 µg C for 2013 samples and $\approx$18 µg C for 2014 and 2015 samples) which is needed for precise $^{14}$C measurements. The diluted samples and blanks are then run through an air processing system at UR that cryogenically dries the air and removes

carbon-containing trace gases other than CO and $CH_4$, followed by CO oxidation over platinized quartz wool at 175˚C ($CH_4$ passes through unaffected), followed by cryogenic trapping and purification of CO-derived $CO_2$. This $CO_2$ is subsequently converted to graphite at the Australian Nuclear Science and Technology Organisation (ANSTO) and $^{14}$C activity is measured on the 10MV ANTARES accelerator mass spectrometer (AMS) (Smith et al., 2010; Tables S1 and S3). Sets of commensurately-sized $^{14}$C standards and blanks are prepared at ANSTO and accompany the samples

through graphitization and measurement. 2 – 3 larger ($\approx$100 µgC) samples of the high-$x_{CO}$ dilution gas were also processed and measured together with each sample set. This allowed to characterize the slight growth of $^{14}$CO in the dilution gas over the years (Tables S1 and S3) resulting from *in situ* cosmogenic $^{14}$CO production in the air cylinder (Lowe et al., 2002).

**2.6 Corrections for procedural effects and calculation of sample [$^{14}$CO]**

The detailed approaches for calculating $^{14}$CO concentration ([$^{14}$CO]) and for correcting for procedural effects have been presented previously (Dyonisius et al., 2020; Hmiel et al., 2020; Petrenko et al., 2016; Petrenko et al., 2013b; Petrenko et al., 2021). For the reader's convenience we also provide a brief description here. [$^{14}$CO] in firn matrix and bubbly ice samples is calculated using:

$$[^{14}_{\phantom{1}}CO] = \frac{pMC}{100} \times e^{-\lambda(y-1950)} \times \frac{\left(1+\frac{\delta^{13}C}{1000}\right)^2}{0.975^2} \times 1.1694 \times 10^{-12} \times x_{CO} \times \frac{1}{22400} \times N_A \times V \qquad (1)$$

where [$^{14}CO$] is the number of $^{14}$CO molecules per g ice, $pMC$ is the sample or blank $^{14}$C activity in percent Modern Carbon (Stuiver and Polach, 1977), $\lambda$ is the $^{14}$C decay constant (1.216 x 10$^{-4}$ yr$^{-1}$), $y$ is the year of measurement, $\delta^{13}C$ is the $\delta^{13}$C of CO in the sample or blank, 0.975 is a factor arising from $^{14}$C activity normalization to $\delta^{13}$C of -25 ‰ associated with pMC, 1.1694 × 10$^{-12}$ is the $^{14}$C / ($^{13}$C + $^{12}$C) ratio corresponding to the absolute international $^{14}$C standard activity (Hippe and Lifton, 2014), 22400 is the number of cm$^3$ STP of gas per mole, $N_A$ is the Avogadro constant and $V$ is the air content in cm$^3$ STP per g ice. For firn air samples and blanks, the air content term does not apply and [$^{14}$CO] is given in units of $^{14}$CO molecules per cm$^3$ STP of air instead.

As the $^{14}$C measurements are made on diluted samples, it is [$^{14}$CO] in the diluted sample air that is initially calculated using $x_{CO}$ and $\delta^{13}$C of CO in diluted samples. The pMC values used for this calculation (Tables S1 and S3) are values that have been empirically corrected for processing effects at ANSTO using measurements of commensurately-sized $^{14}$C standards and blanks (Petrenko et al., 2021); this correction has a 0 – 2 % effect for samples in this study. The calculation is then repeated to determine the [$^{14}$CO] contribution in the diluted samples that is due to the high-$x_{CO}$ low-$^{14}$C dilution gas and this contribution is then subtracted from the total. The remaining [$^{14}$CO] is further corrected for the effect of air dilution associated with adding the dilution gas to the samples (Petrenko et al., 2021; Tables S1 and S4).

The field procedural blanks collected during each season (Tables S1 – S4) characterize all or most of the extraneous [$^{14}$CO] in the samples, which is mainly due to *in situ* $^{14}$CO production in the canisters containing sample air during storage and transport prior to sample processing. For 2013 firn air samples, the procedural blanks are fully representative of all extraneous $^{14}$CO in the samples. However, the blanks were collected 11 days apart, and procedural blank 1 shows significantly higher [$^{14}$CO] due to the 11 extra days of *in situ* $^{14}$CO production in the canister during storage at the high-altitude Summit site (Table S1). The procedural blank correction for the 2013 samples therefore accounted for differences in exposure time at Summit based on sample collection date and *in situ* $^{14}$CO growth rate at Summit as determined from the two procedural blanks. Fully corrected firn air sample $^{14}$CO values are shown in Table S1 and Figure 3.

For 2014 and 2015 samples, several corrections are needed beyond the dilution correction. First, [$^{14}$CO] for both samples and blanks is corrected for the effect of gas dissolution in the ice melter (<1% effect; Table S4) using solubility equilibrium parameters determined from δXe/Kr. The samples are next corrected for extraneous [$^{14}$CO] as characterized by procedural blanks from the same field campaign (Table S4). For 2014, the same standard gas was used in all sample and blank extractions, and this gas was not passed through Sofnocat to remove $^{14}$CO. For this set of samples, there was no significant difference in [$^{14}$CO] between procedural blank 1 (start of season) and procedural

blank 2 (end of season). This is because [14]CO was being produced *in situ* in the sample and blank canisters at the same rate as it was being produced in the 2014 standard gas cylinder. For this reason, for 2014 samples the average [[14]CO] for procedural blanks was subtracted from sample [[14]CO]. For 2015, the standard gases used for samples and blanks were passed through Sofnocat, removing all [14]CO. Because of this, the blank correction for 2015 samples accounted for differences in exposure time at Summit using the same approach as for 2013 samples.

There are two additional sources of extraneous [14]CO in the 2014 and 2015 samples that the field procedural blanks do not characterize. First, the field procedural blanks do not properly mimic heating of the ice melter walls and associated CO outgassing. Second, $x_{CO}$ in trapped air in Greenland ice is known to be affected by (non-cosmogenic) *in situ* production in the ice likely originating from organic impurities (Fain et al., 2014; Fain et al., 2022). Both the $x_{CO}$ contribution and the [14]C activity of this extraneous CO are needed to estimate the contribution to [[14]CO]. This extraneous $x_{CO}$ contribution for firn matrix samples (Table S3) is calculated by subtracting the average $x_{CO}$ of the field procedural blanks for the same season from sample $x_{CO}$. For LIZ and bubbly ice samples (which do contain trapped air), we use a northern hemisphere high latitude $x_{CO}$ history compiled by Hmiel et al. (2020) based on direct atmospheric observations as well as firn air and ice core measurements (Haan and Raynaud, 1998; Petrenko et al., 2013a and references therein) to predict expected $x_{CO}$ in trapped air. For bubbly ice, the additional extraneous $x_{CO}$ contribution from outgassing from hot melter walls and *in situ* CO production in the ice (Table S3) is estimated as follows:

$$x_{CO,extraneous} = x_{CO,measured} - x_{CO,2015\ field\ blank\ average} - x_{CO,expected\ from\ trapped\ air} \qquad (2)$$

For LIZ samples, the calculation is similar but with the expected $x_{CO}$ from trapped air estimated based on the $x_{CO}$ atmospheric history and the fraction of air in the sample arising from air bubbles as determined from δNe/N$_2$ measurements (more trapped air lowers the Ne/N$_2$ ratio in these samples that used a high-Ne standard gas to supplement air content). The [14]C activity of the extraneous CO is uncertain; we thus use a value of 50 pMC and a 2σ uncertainty of 50 pMC, allowing for the full range from [14]C-free to [14]C-modern. The relative effect of the correction for this extraneous [14]CO is large for the surface sample (39%), small for bubbly ice samples (2 - 3%) and intermediate (3 - 11%) for other samples; this correction contributes significantly to overall uncertainty (Table S4).

Shallow firn above the LIZ contains a small amount of trapped air, possibly in microbubbles (e.g., Siegenthaler et al., 2005). As this microbubble air is not well understood and is not included in the closed porosity parameterization we use in our model (Section 3.1), we further correct [[14]CO] in the firn matrix samples for microbubble air. The amount of microbubble air is estimated from the comparison of measured δXe/Ar and δKr/Ar values with those expected from the solubility equilibrium in the ice melter (addition of trapped air lowers the Xe/Ar and Kr/Ar ratios in these samples that use a high-Kr, high-Xe gas as a carrier). The magnitude of the correction for [14]CO from microbubbles is 0.6 – 2% (Table S4). As a final step in the calculation for 2014 and 2015 samples, the fully-corrected [[14]CO] (in units of [14]CO molecules / cm$^3$ STP) is converted to [14]CO content per gram of ice (Table S4; Figure 3) via multiplying by the effective air content in the samples ($V$ in Equation 1; in cm$^3$ STP / g ice; Table S2). For the firn matrix and LIZ samples, the air content is determined based on the sampled ice mass and recovered amount of air. For bubbly ice samples, the air content is taken as 0.0904 cm$^3$ STP / g ice, which is the average value for the

last 2000 years from the nearby GRIP ice core (Raynaud et al., 1997). The firn model is tuned to match the same air content in ice below the LIZ.

## 3 Model characterization of firn and ice $^{14}$CO

### 3.1 Overview of the firn model used for the Greenland Summit site

Interpretation of $^{14}$CO results is done using an established firn gas transport model – the CIC model from Buizert et al. (2012), which has been modified for this study to include production and movement of in situ $^{14}$C (Section 3.2); herein referred to as the "firn model". The model assumes a steady-state, isothermal firn column with an accumulation rate 0.235 m ice equivalent yr$^{-1}$, temperature of $-31\degree$C, firn densities based on a fit to the Summit density measurements by Adolph and Albert (2014), and a vertical diffusivity profile that is calibrated using firn air measurements of several traces gases with known atmospheric histories. The model simulates transport of trace gas mole fractions in the open and closed porosity (air bubbles) and can extend to depths below the firn zone. The total air content in the firn model below close-off (where all the air has been trapped into bubbles) is tuned to match the value of 0.0904 cm$^3$ STP / g ice from Raynaud et al. (1997) as mentioned above.

The method for tuning the depth-diffusivity profile is as described in the supplement to Buizert et al. (2012), with a few modifications for the Summit site as follows. (1) $^{14}$CO$_2$ was not used as a tracer gas due to uncertainty about the magnitude of in situ cosmogenic $^{14}$CO$_2$ production. (2) $\delta^{15}$N$_2$ was also not used due to observed thermal fractionation signal from recent warming and seasonal temperature gradients (e.g., Severinghaus et al., 2001) that are not well-captured by the steady-state model. (3) N$_2$O was included as a tracer gas using a recently developed multi-site reconstruction from firn air (Prokopiou et al., 2017). (4) All northern hemisphere atmospheric histories for trace gases used in model tuning were updated through December 2015 with monthly mean flask measurements from Summit Station by NOAA ESRL (Dlugokencky et al., 2018; Petron et al., 2018) and converted to the most recent NOAA measurement scales (Hall et al., 2014).

Firn gas transport models consider layer thinning by densification, but do not normally consider additional ice thinning related to deformation by ice flow, because such thinning effects are typically very small in the firn column. As our measurements extend to ≈130 m, however, the effects of flow-related thinning do need to be considered for the deepest samples. Measurements that could provide a complete ice age - depth scale (and a complete empirical thinning function) are unfortunately not available at our site. However, Hmiel et al. (2020; see their Supplement) used age tie points from continuous flow chemistry analyses on sections of ice cores from our site between 67 – 98 m depth to show that a constant accumulation rate of 0.235 m ice equivalent yr$^{-1}$ in the model yields an excellent match to these tie points. Hmiel et al. (2020) further showed that these same depth-age tie points can be matched well in a Dansgaard-Johnsen (DJ) thinning model (Dansgaard and Johnsen, 1969) that uses a constant accumulation rate of 0.246 m ice equivalent yr$^{-1}$. For the purposes of the firn model, we use the accumulation rate of 0.235 m ice equivalent yr$^{-1}$, as this also yields a good match to trace gas measurements. However, we apply a small depth correction for ice deeper than 112 m, where the depth – age scales for the scenario with constant accumulation

rate of 0.235 m ice equivalent yr$^{-1}$ and the DJ model scenario with 0.246 m ice equivalent yr$^{-1}$ start to diverge significantly. This correction adjusts the depths used for *in situ* $^{14}$CO calculations to those predicted by the DJ model scenario, resulting in a 1.25 m shallower depth in the model for the range of the deepest (130 m) sample and a ≈1% increase in model-calculated $^{14}$CO content for that sample.

### 3.2 Parameterization of *in situ* cosmogenic $^{14}$CO in the firn model

The firn model accounts for *in situ* $^{14}$C production in the ice grains by secondary cosmic ray neutrons and muons, for the fraction of this $^{14}$C that forms $^{14}$CO as well as for any loss of this $^{14}$C from the ice grains via leakage into the open porosity (firn air) or closed porosity (air bubbles). Figure 1 illustrates $^{14}$CO production rates by secondary cosmic ray neutrons and muons versus depth at Summit, and the following sections describe how *in situ* cosmogenic $^{14}$CO was implemented in the model for the purposes of this study, with an overview of the relevant model parameters presented in Table 1.

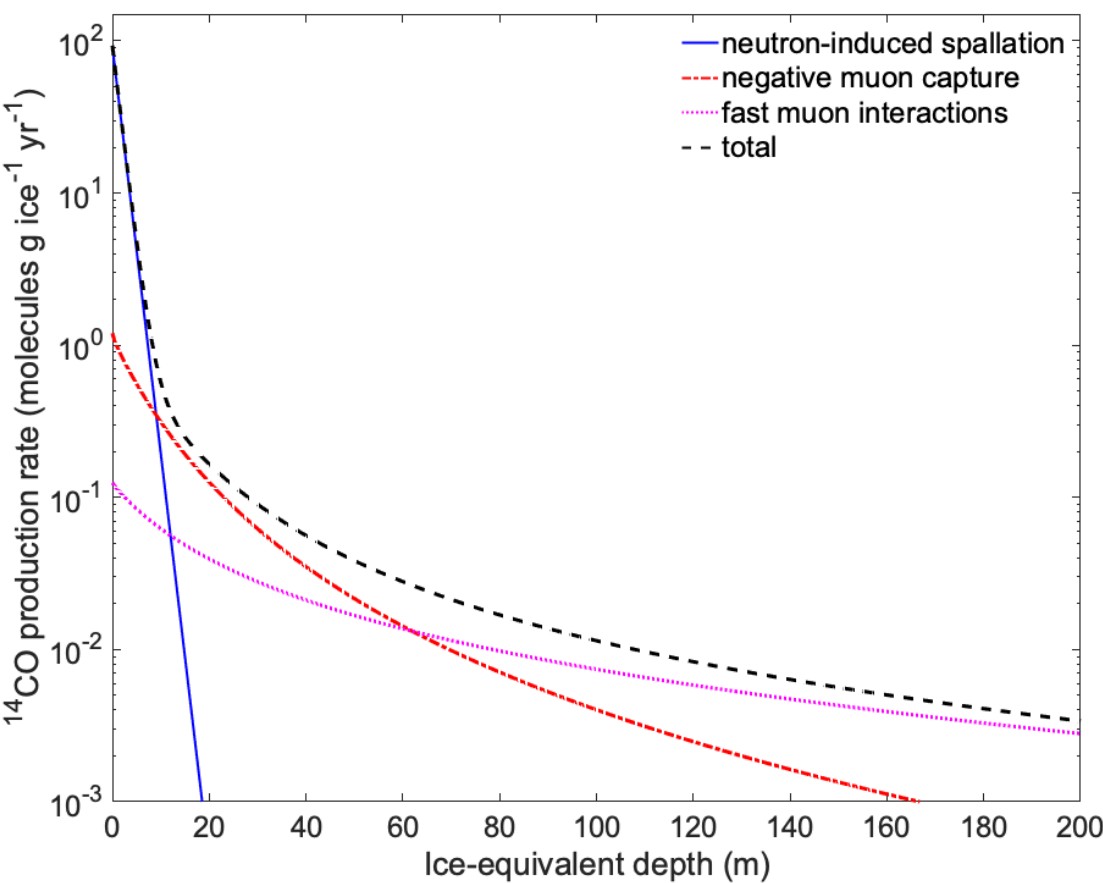

**Figure 1: Predicted $^{14}$CO production rate vs depth.** $^{14}$CO production rates vs depth in ice by the neutron and muon mechanisms, calculated for the Greenland Summit location. Production rate from neutrons is calculated using

Equation 3 as described in Section 3.2.1. Production rates for the muon mechanisms are calculated using Equations 6 and 7 as described in Section 3.2.2, using best-estimate $f_\mu$ values from Dyonisius et al. (2023).

### 3.2.1 $^{14}CO$ production by neutrons

*In situ* $^{14}C$ production rate by neutron-induced spallation of $^{16}O$ has been well constrained from measurements in near-surface quartz samples (Borchers et al., 2016; Young et al., 2014); these $^{14}C$ production rates for a sea level high latitude (SLHL) site ($P_{n,SLHL}^{Qtz}(0)$) can be translated to ice ($P_{n,SLHL}^{Ice}(0)$) by accounting for the difference in oxygen

atom density (atoms/g) between quartz and ice ($Q_c$, factor of 1.667). The subscript *n* denotes the neutron mechanism

| Model parameter | Description | Source | Uncertainty |
|---|---|---|---|
| $P_{n,SLHL}^{Qtz}(0)$ | $^{14}C$ production rate by neutrons in quartz at the surface at sea level and high latitude | Borchers et al. (2016) | 10% (included in model via the $F_n$ parameter) |
| $S_{n,sum}$ | scaling factor for $^{14}C$ production by neutrons at Greenland Summit | Calculated using Lifton et al. (2014) scaling model | Not included |
| $Q_c$ | difference in oxygen atom density between quartz and ice | Calculated in this study | Not included |
| $\Lambda_n$ | absorption mean free path of neutrons in ice | Lal et al. (1987) | Not included |
| $\Omega^{CO}$ | fraction of total *in situ* $^{14}C$ in ice that forms $^{14}CO$ | Van der Kemp et al. (2002); Dyonisius et al. (2023) | Not included for neutrons; for muons is folded into $f_{\mu-}$ and $f_{\mu f}$ |
| $F_n$ | adjustable parameter that accounts for uncertainty in $P_{n,SLHL}^{Qtz}(0)$ during grid search approach | Specified in model | Not applicable |
| $R_1$ | fraction of *in situ* $^{14}CO$ that is initially retained in the ice grains | Determined in this study via grid search approach | |
| $L_1$ | fraction of *in situ* $^{14}CO$ in the ice grains that leaks out per year | Determined in this study via grid search approach | |
| $R_{\mu-}(h)$ | stopping rate of negative muons at mass-depth $h$ | Balco et al. (2008) | Not included |
| $f_C$ | probability that the negative muon is captured by $^{16}O$ (chemical compound factor) | Heisinger et al. (2002a) | Not included |
| $f_D$ | probability that the negative muon does not decay prior to nuclear capture | Heisinger et al. (2002a) | Not included |
| $f^*$ | probability for $^{14}C$ production from $^{16}O$ after negative muon capture | Heisinger et al. (2002a) | Not included |
| $\sigma_0$ | reference nuclear reaction cross section for $^{14}C$ production from $^{16}O$ at muon energy of 1 GeV | Heisinger et al. (2002b) | Not included |
| $\beta(h)$ | unitless depth dependence factor | Balco et al. (2008) | Not included |
| $\phi(h)$ | total muon flux at mass-depth $h$ | Balco et al. (2008) | Not included |
| $\bar{E}(h)$ | mean muon energy at mass-depth $h$ | Balco et al. (2008) | Not included |
| $\alpha$ | power factor that describes the energy dependence of the cross section for fast muon mechanism | Heisinger et al. (2002b) | Not included |
| $N$ | number of target nuclei ($^{16}O$) per gram ice | Calculated in this study | Not included |
| $f_{\mu-}$ | Tuning factor (negative muon mechanism) for $^{14}CO$ production rate | Determined in this study via trialing range of values from Dyonisius et al. (2023) | |
| $f_{\mu f}$ | Tuning factor (fast muon mechanism) for $^{14}CO$ production rate | Determined in this study via trialing range of values from Dyonisius et al. (2023) | |

| [14CO] history | Atmospheric [14CO] history over Greenland Summit | This study | See section 4.3.2 |

Table 1. Summary of model parameters involved in calculations of [14CO].

for [14C] production. Surface production rates at different locations are estimated as a function of surface pressure and geomagnetic latitude using established 'scaling models' (e.g., Desilets et al., 2006; Lifton et al., 2014; Stone, 2000) which use measurements and / or models of the secondary cosmic ray flux in the atmosphere. In this study, we use the scaling model of Lifton et al. (2014) for neutron-induced [14C] production to determine the scaling factor for our site ($S_{n,Sum}$). The SLHL production rate of 12.76 molecules / g Qtz / yr from the CRONUS-Earth project (Borchers et al., 2016) is used for $P_{n,SLHL}^{Qtz}(0)$. This value was chosen as it was calculated using the same scaling model (Lifton et al., 2014) as used in this study; production rate differences between scaling models are on the order of ±10% and generally agree within measurement uncertainty for high latitude locations.

We use a value of $\Omega^{CO} = 0.31$ for the fraction of total *in situ* [14C] in ice that forms [14CO] (Dyonisius et al., 2023; van der Kemp et al., 2002). We note that this value is based on prior measurements in ice where part or all of the *in situ* [14C] originated from production via the muon mechanisms. However, since all newly-produced "hot" *in situ* [14C] atoms have to lose most of their energy prior to reacting (regardless of production mechanism), there is no a priori reason why the partitioning of *in situ* [14C] would be different for different production mechanisms.

[14C] production from neutrons declines exponentially with increasing mass-depth ($h$, in g cm$^{-2}$); we use a value of 150 g cm$^{-2}$ for the absorption mean free path $\Lambda_n$ of neutrons in ice (e.g., Lal et al., 1987; van de Wal et al., 2007). The model also includes an adjustable parameter $F_n$ that allows for tuning the model production rate within uncertainties. These parameters are combined to calculate the [14CO] production rate by neutrons (in [14CO] molecules / g ice / yr) as a function of mass-depth $h$:

$$P_n^{CO}(h) = \Omega^{CO} \cdot F_n \cdot S_{n,Sum} \cdot Q_C \cdot P_{n,SLHL}^{Qtz}(0) \cdot e^{-h/\Lambda_n} \qquad (3)$$

As can be seen on Figure 1, [14CO] production by neutrons is only significant in the top ≈10 m ice equivalent depth. Neutron fluxes at high latitude sites are modulated by the heliospheric magnetic field, and over the range of ice ages (≈500 years) that correspond to the sampled depth range (≈ 0 – 130 m), the Lifton et al. (2014) model predicts variability in $P_n(0)$ at Summit of up to ≈25%. However, for deeper ice samples (which would have been most affected by the temporal $P_n(0)$ change), the neutron-produced contribution to total [14CO] is 20% or less. As temporal variations in $P_n(0)$ have a much smaller effect on total [14CO] than uncertainties in other parameters such as muogenic production rates and the atmospheric [14CO] history, we assume no temporal variability in $P_n(0)$ for the purposes of this study.

### 3.2.2 [14CO] production by muons

For *in situ* [14C] production from [16O] via the muon mechanisms, we use the parameterizations developed by Heisinger et al. (2002a) and Heisinger et al. (2002b) as implemented in Matlab by Balco et al. (2008) (also Model 1A

in Balco, 2017); herein referred to as the "Balco model", with all relevant parameters adjusted for ice. Briefly, the Heisinger et al. (2002a) production rate (in $^{14}$C atoms g$^{-1}$ yr$^{-1}$) parameterization for the negative muon capture mechanism is:

$$P_{\mu-}(h) = R_{\mu-}(h) \cdot f_C \cdot f_D \cdot f^*$$  (4)

where $R_{\mu-}(h)$ is the stopping rate of negative muons (muons g$^{-1}$ yr$^{-1}$) at mass-depth $h$, $f_C$ is the chemical compound factor representing the probability that the stopped muon is captured by one of the target atoms, $f_D$ is the probability that the negative muon does not decay in the K-shell before nuclear capture, and $f^*$ is the effective probability for production of the cosmogenic nuclide of interest after $\mu^-$ capture by the target nucleus. We use the $f$ values given by Heisinger et al. (2002a) for production of $^{14}$C from $^{16}$O in ice.

Heisinger et al. (2002b) parameterizes production rate from fast muons as:

$$P_{\mu f}(h) = \sigma_0 \cdot \beta(h) \cdot \phi(h) \cdot \bar{E}(h)^\alpha \cdot N$$  (5)

where $\sigma_0$ is the reference nuclear reaction cross section at muon energy of 1 GeV (cm$^2$), $\phi(h)$ is the total muon flux at mass-depth $h$ (muons cm$^{-2}$ yr$^{-1}$ sr$^{-1}$), $\beta(h)$ is a unitless mass-depth dependence factor, $\bar{E}(h)$ is the mean muon energy at mass-depth $h$ (GeV), $\alpha$ is a power factor that describes the energy dependence of the cross section (we use $\alpha=0.75$, consistent with Dyonisius et al., 2023 and Heisinger et al., 2002b), and N is the number of target nuclei per gram target mineral.

The above muogenic production rate parameterizations are incorporated within the Balco model, which also provides the needed altitude scaling of muon fluxes and energies based on atmospheric pressure at Summit. The $^{14}$CO production rate via the negative muon and fast muon mechanisms as a function of mass-depth $h$ is then calculated as:

$$P_{\mu-}^{CO}(h) = f_{\mu-} \cdot P_{\mu-}^{Balco}(h, P)$$  (6)

$$P_{\mu f}^{CO}(h) = f_{\mu f} \cdot P_{\mu f}^{Balco}(h, P)$$  (7)

where $P_{\mu}^{Balco}(h, P)$ is the total $^{14}$C production rate (in atoms g$^{-1}$ yr$^{-1}$) from the respective muon mechanism at mass-depth $h$ and surface pressure $P$ and $f_{\mu-}$ and $f_{\mu f}$ are dimensionless tuning factors that account for 1) the fraction of total $^{14}$C that forms $^{14}$CO ($\Omega^{CO}$) and 2) adjustment factor for production rate. This definition of $f_{\mu-}$ and $f_{\mu f}$ is consistent with

the Dyonisius et al. (2023) study of muogenic $^{14}$CO production rates in ice at Taylor Glacier, Antarctica. Figure 1 illustrates the predicted muogenic $^{14}$CO production rates at a range of depths at Summit.

Only muons with sufficiently high energy can penetrate the Summit firn column to the depths (70 – 80 m) where the first impermeable ice layers start to form and *in situ* $^{14}$C retention in the ice / firn starts to increase. The primary cosmic rays responsible for production of such higher-energy muons are sufficiently high-energy to be

insensitive to solar modulation (Petrenko et al., 2024). We therefore do not assume any temporal variability in muon fluxes.

### 3.2.3 $^{14}$C retention, leakage and decay

The firn model defines two reservoirs within the ice grains to contain the *in situ* produced $^{14}$C (Fig. 2). The

partitioning of $^{14}$C between these reservoirs is numerically defined by coefficients $R_0$ and $R_1$, representing the fraction of *in situ* produced $^{14}$C entering each reservoir. For each reservoir, the model also defines leakage coefficients $L_0$ and

$L_1$ which represent the fraction of total *in situ* [14]C that leaks out of the respective reservoir per year (for $L_1$) or per time step (for $L_0$). We introduce these two reservoirs because a preliminary analysis showed that using a single ice grain reservoir does not provide a good fit to the observations; physical justification for these two reservoirs is discussed in Section 4.2. The relative partitioning of leaked [14]C between open and closed porosity scales with the relative volumes of open and closed porosity in the model at a given depth, such that the fraction leaked into the open pores equals $L_i \times \frac{s_{op}}{s}$ and the fraction leaked into the closed pores $L_i \times \frac{s_{cl}}{s}$, where $s_{op}$ and $s_{cl}$ are the open and closed porosity, respectively, and the total porosity $s = s_{op} + s_{cl}$ (Fig. 2). Any [14]CO that leaks into the closed porosity is retained permanently, and [14]CO that leaks into the open pores can escape the firn via upward diffusion into the atmosphere.

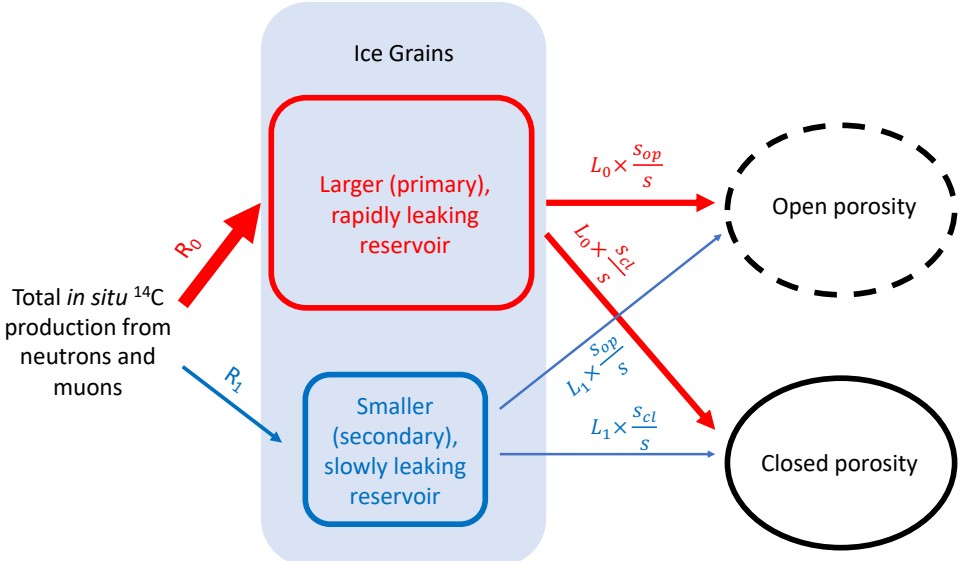

**Figure 2: Conceptual diagram of model parameterization of *in situ* [14]C partitioning between ice grain reservoirs and leakage into porosity.**

Prior studies have found that the majority of *in situ* cosmogenic [14]C produced in the upper firn column is lost to the atmosphere and not retained through firn densification into ice (e.g., de Jong et al., 2004; Petrenko et al., 2013b; Smith et al., 2000). The model accounts for the rapid loss of most [14]C from ice grains by defining the leakage coefficient $L_0$ of the primary reservoir so that all [14]C in this reservoir is lost to porosity at every model time step (time step = 0.5 yr) . In tuning the model, we find that the partitioning coefficient $R_0$ is close to 1, which means that for a given model timestep the majority of the *in situ* [14]C produced at all depths is sorted into the primary reservoir, and all of it leaks from the ice grains into the porosity. In the upper firn, which is dominated by open porosity, this [14]C typically escapes to the atmosphere, whereas in the LIZ the *in situ* [14]C that leaks from the grains into the open and closed porosity is inhibited from exchange with the atmosphere due to limited vertical diffusion. For ice below bubble close off (~80m at Summit), there is no longer any open porosity in the model and thus any *in situ* [14]C that leaks out of the ice grains must leak into the closed porosity.

This parameterization enables tuning of the *in situ* [14]C content in the firn matrix to match the observations by specifying the secondary reservoir partitioning and leakage coefficients ($R_1$, $L_1$) while the primary reservoir partitioning coefficient is determined from the conservation of mass ($R_0 = 1 – R_1$) and its leakage coefficient $L_0$ is fixed as described above.

Radioactive decay of [14]C is considered using the standard exponential decay equation with a decay constant
of $1.210 \times 10^{-4}$ yr$^{-1}$, corresponding to a [14]C half-life of 5730 years (Godwin, 1962). Finally, the downward advection of ice is considered at each time step by adding a new ice parcel (with no *in situ* [14]C content) at the uppermost box of the model, shifting each ice parcel downwards by one box of the depth resolution, and removing the bottommost ice parcel.

As is described in Section 4 below, the firn model is used in combination with the measurements to constrain
the possible ranges of four parameters relevant to production and retention of [14]CO in the firn and ice ($R_1$, $L_1$, $f_{\mu-}$ and $f_{\mu f}$). While the model has a large number of parameters, we note that for most of these parameters the values are either available from prior studies or determined outside of the model (Table 1). We note further that we only constrain / tune two model parameters at once ([$R_1$, $L_1$] or [$f_{\mu-}$ , $f_{\mu f}$]).

## 4 Results and Discussion

### 4.1 Main features of results and overview of approach for constraining *in situ* [14]CO parameters

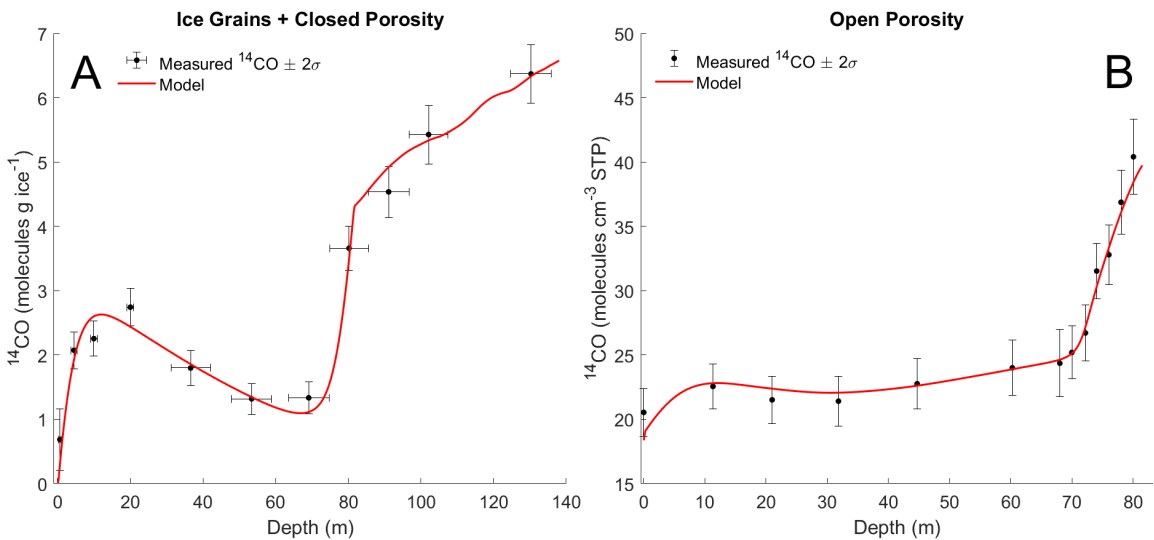

**Figure 3: [14]CO measurements at Summit.** [[14]CO] measured at Summit in ice grains + closed porosity (panel A) and in open porosity (firn air; panel B), together with an example of our firn model fit to the data. Measured values are after all corrections (Table S4).

Figure 3 shows the measurements after all corrections in ice grains + closed porosity (Fig. 3a) and in firn air (Fig. 3b) together with an example of a firn model fit. First, we discuss [[14]CO] in the ice grains and closed porosity

(Fig. 3a). In the shallow firn (0 – 60 m), the amount of closed porosity is very small and both the measurements and the model represent [$^{14}$CO] in the ice grains only. [$^{14}$CO] first increases rapidly with depth to a peak in the 10 – 20 m depth range, mainly due to $^{14}$CO production by the neutron mechanism. [$^{14}$CO] then declines gradually with depth through the shallow firn to a minimum at ≈60 m. As firn layers move downwards via the processes of continued snow accumulation at the surface, densification and ice flow, the layers carry with them the ice grain $^{14}$CO content they acquired above. Further, $^{14}$CO production continues (by muons) at intermediate firn depths (20 – 60 m). Therefore, the fact that $^{14}$CO content in the ice grains decreases rather than increases with increasing depth between 20 and 60m indicates slow $^{14}$CO leakage out of the ice grains. $^{14}$C radioactive decay is far too slow to explain this, as the ice layers traverse the entire firn column at Greenland Summit in only ≈200 years. [$^{14}$CO] then increases rapidly with depth in the LIZ (≈69 – 80 m), mainly reflecting the air trapping process that moves $^{14}$CO from the open into the closed pores. Below ≈80 m, no further air trapping takes place (air is already all in trapped bubbles) and the continued more gradual increase of [$^{14}$CO] with depth is due to production by deep-penetrating muons.

Second, we look at the open porosity [$^{14}$CO] profile (Fig. 3b). It shows a small peak at ≈10 m, representing the previous winter's seasonal maximum in atmospheric [$^{14}$CO]. Firn air [$^{14}$CO] increases gradually with depth between 30 – 70 m. This reflects the fact that at deeper levels in the firn, the diffusive gas exchange with the atmosphere is relatively slower, allowing some of the $^{14}$CO produced by the muon mechanisms to accumulate. Firn air [$^{14}$CO] increases rapidly in the LIZ, as the impermeable ice layers present in the LIZ almost completely stop vertical gas diffusion, trapping all $^{14}$CO that is produced by muons at these depth levels.

The different tuning parameters in the firn model are each constrained most strongly by different depth ranges of the [$^{14}$CO] profile. Because firn shallower than ≈60 m contains almost no closed porosity, [$^{14}$CO] in the firn matrix in this depth range is sensitive only to *in situ* production, retention and leakage rates. Further, production rates in the shallowest firn (≈0 – 10 m) are 1 – 2 orders of magnitude higher than rates in mid-depth firn (≈40 m), and in the shallowest firn production is dominated by the neutron mechanism (Fig. 1). As neutron $^{14}$C production rates as well as the $^{14}$CO fraction of total $^{14}$C are well constrained (Section 3.2.1), this allows us to use measurements in the 0 – 60 m depth range to constrain the key relevant parameters in the firn model: $R_1$, the fraction of produced $^{14}$CO that initially stays in the ice grains and $L_1$, the leakage rate of this initially retained $^{14}$CO into porosity (Fig. 2). As mentioned above, $R_0 = 1 – R_1$ and $L_0 = 1$, so these parameters are not tuned in the model.

[$^{14}$CO] in LIZ firn air and in bubbly ice below the LIZ is mainly sensitive to muogenic production rates, and to a lesser extent the atmospheric [$^{14}$CO] history. We therefore use the [$^{14}$CO] measurements in this depth range to provide improved constraints on the muogenic $^{14}$CO production rates. Model [$^{14}$CO] in ice grains and closed porosity in the LIZ is extremely sensitive to model characterization of the air bubble trapping process. As this characterization is imperfect, we do not use firn/ice samples from the LIZ (70 and 80m samples in Tables S2 – S4) for constraining *in situ* $^{14}$CO parameters.

## 4.2 Retention and leakage of *in situ* cosmogenic $^{14}$CO in the firn

To constrain the best-estimate and possible range of values of the $R_1$ and $L_1$ parameters we employ a grid search approach that involves repeated runs of the firn model while varying these parameters within reasonable ranges

determined from preliminary manual tuning of the model to observations. In addition to varying $R_1$ and $L_1$, the grid search also varies $F_n$ (the adjustment factor for production rate from neutrons; Section 3.2.1). The grid search scans over values for $F_n$ ranging from 0.9 to 1.1 at an interval of 0.01, for $R_1$ ranging from 0.3% to 0.7% at an interval of 0.02%, and for $L_1$ ranging from 0.3% $yr^{-1}$ to 1.3% $yr^{-1}$ at an interval of 0.05%. For the purposes of this grid search, the muogenic $^{14}CO$ production rate adjustment factors $f_{\mu-}$ and $f_{\mu f}$ are 0.068 and 0.085 respectively, following the best-

estimate values from Taylor Glacier measurements (Dyonisius et al., 2023). We also trialed lower $f_{\mu-}$ and $f_{\mu f}$ values of 0.0584 and 0.0633 that are more consistent with $^{14}CO$ observations in ice below the LIZ (Section 4.3); this did not affect the results of this grid search for optimization of $R_1$ and $L_1$ parameters. In total, the grid search amounts to all permutations of 21 different values for each of $F_n$, $L_1$ and $R_1$, resulting in $21^3 = 9261$ simulations.

The best-estimate set of parameters minimizes the model-data mismatch for the firn matrix samples (0 – 60

480 m depth range; mismatch defined by the reduced $\chi^2_\nu$ value) for the case where $F_n = 1$ (Fig. 4a, purple trace). The best-estimate value for $R_1$ is 0.46% and the best-estimate value for $L_1$ is 0.60 % $yr^{-1}$. To define an uncertainty range of these parameters, we accept all solutions from the grid search where the model calculates $^{14}CO$ within a $2\sigma$ bound of the best solution. The $2\sigma$ bound is the mean $2\sigma$ analytical uncertainty for the firn matrix $^{14}CO$ measurements and the range of these accepted solutions is represented by the blue shading in Figure 4a. The contour of accepted solutions

in the $R_1$, $L_1$ space is shown on Figure 4b. We also determined which combinations of parameters (from the accepted set of solutions) result in the minimum and maximum $^{14}CO$ content in ice below the LIZ, as this is important for constraining $f_{\mu-}$ and $f_{\mu f}$ values (Section 4.3). The results of the grid search are summarized in Table 2.

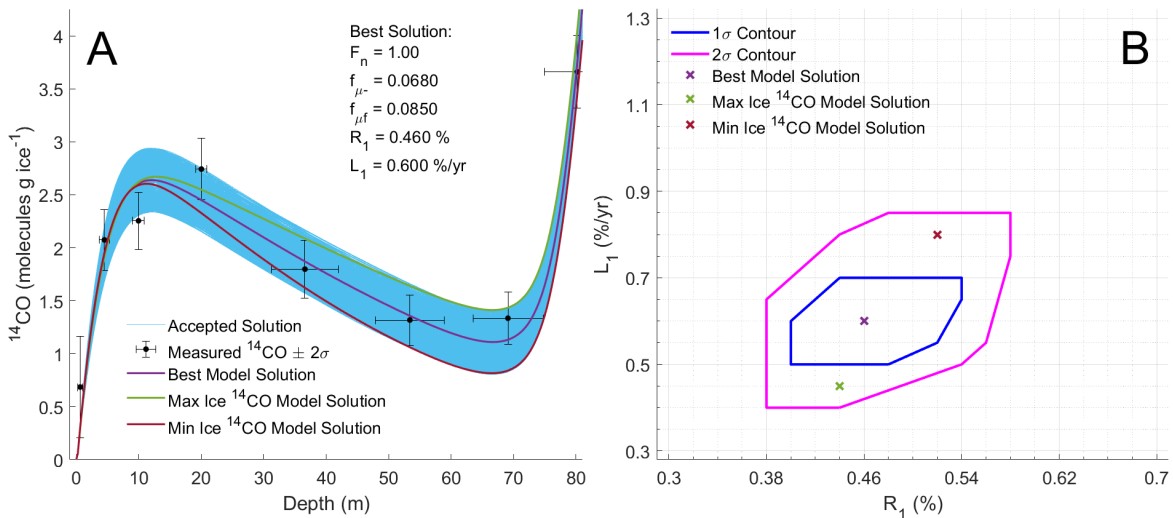

**Figure 4: Constraints on $^{14}CO$ retention and leakage in Summit firn.** Panel A shows results for the firn matrix

samples that were used to provide the constraints, together with the set of accepted solutions (blue envelope), as well as the best-fit (minimum $\chi^2$) solution and the accepted solutions that yield minimum ("Min Ice") or maximum ("Max Ice") $^{14}CO$ content in ice below the LIZ. Panel B shows the set of accepted solutions as well as the best, Min Ice and Max Ice solutions in $R_1 - L_1$ space. $R_1$ is the fraction of produced $^{14}CO$ that initially stays in the ice grains and $L_1$ is the leakage rate of this initially retained $^{14}CO$.

| Parameter | Best-estimate value | $2\sigma$ range | "Min": Set of values yielding minimum [14]CO in ice below LIZ | "Max": Set of values yielding maximum [14]CO in ice below LIZ |
|---|---|---|---|---|
| $F_n$ | 1.00 | 0.9 – 1.1 | 0.9 | 1.03 |
| $R_1$ (%) | 0.46 | 0.38 – 0.58 | 0.52 | 0.44 |
| $L_1$ (% yr$^{-1}$) | 0.60 | 0.40 – 0.85 | 0.80 | 0.45 |

**Table 2. Results of the grid search for R$_1$ and L$_1$ parameters in the firn matrix.** Note that the values listed for F$_n$ under the 2$\sigma$ range column are not determined from the grid search but rather assumed based on published uncertainty estimates for $^{14}$C production rates by neutrons (Section 3.2.1.)

Previous discussions of low retention of *in situ* cosmogenic $^{14}$C in the firn focused on mechanisms such as firn sublimation and recrystallization in combination with wind ventilation (e.g., Lal et al., 2001; Petrenko et al., 2013b). However, our results are quantitatively consistent with a mechanism that involves a very rapid (< 1 year) loss from the ice grains of >99% of produced $^{14}$C; too fast to be explained by recrystallization or ventilation. The observation of rapid loss of almost all *in situ* $^{14}$C from the ice grains is also consistent with the results of *in situ* $^{14}$C production rate comparison between the Antarctic blue ice areas of Taylor Glacier (Dyonisius et al., 2023; Petrenko et al., 2016) and Scharffenbergbotnen (van der Kemp et al., 2002) that Dyonisius et al. (2023) conducted. The Taylor Glacier results included $^{14}$CO measurements that were obtained with the same method as this study as well as $^{14}$CO$_2$ measurements that used an ice sublimation technique, while the Scharffenbergbotnen $^{14}$CO and $^{14}$CO$_2$ measurements were made using a dry extraction technique. There has been concern with dry-extraction measurements with regard to incomplete release of *in situ* $^{14}$C from the ice grains (Smith et al., 2000; van der Kemp et al., 2002). However, if almost all *in situ* $^{14}$C is rapidly transferred to the air porosity after production, then the dry extraction technique would be expected to give equivalent results.

We hypothesize that the rapid loss of *in situ* $^{14}$C from ice grains is mainly driven by diffusion of the $^{14}$C-containing gas molecules ($^{14}$CO, $^{14}$CH$_4$, $^{14}$CO$_2$). While we are not aware of prior estimates of CO diffusivity through ice, both measurement-based estimates and estimates based on molecular dynamics simulations indicate that gases with smaller molecular diameters diffuse faster through ice than gases with larger molecular diameters (e.g., Patterson and Saltzman, 2021 and references therein). There are estimates of gas diffusivity through ice available for CH$_4$ (Ikeda-Fukazawa et al., 2004; Noguchi et al., 2019) and O$_2$ (Ikeda-Fukazawa et al., 2005). CH$_4$ has a slightly larger molecular diameter than CO, while O$_2$ has a slightly smaller molecular diameter (Severinghaus and Battle, 2006); thus diffusivity estimates for CH$_4$ and O$_2$ should be indicative of the possible range for CO diffusivity through ice. Noguchi et al. (2019) experimentally determined the diffusivity of CH$_4$ through ice at 257 K, with $D_{CH4,257K} = (5.2 \pm 1.4) \times 10^{-11}$ m$^2$ s$^{-1}$. Ikeda-Fukazawa et al. (2004) used molecular dynamics simulations to estimate CH$_4$ diffusivity, with average $D_{CH4,257K} = 3.6 \times 10^{-11}$ m$^2$ s$^{-1}$, in fairly good agreement with the experimental estimate. To be conservative, we take the lower estimate (Ikeda-Fukazawa et al., 2004) and recalculate it (following the temperature dependence in Ikeda-

Fukazawa et al., 2004) for the temperature of Greenland Summit (-31°C; Buizert, 2013). This yields $D_{CH4,242K} = 1.2 \times 10^{-11}$ m$^2$ s$^{-1}$.

For $O_2$, Ikeda-Fukazawa et al. (2005) used molecular dynamics simulations to estimate the diffusivity through ice as a function of temperature. This relationship yields $D_{O2,242K} = 2.8 \times 10^{-11}$ m$^2$ s$^{-1}$. The Ikeda-Fukazawa et al. (2005) estimates were recently called into question by Oyabu et al. (2021) who used observations of diffusive smoothing of the $\delta O_2/N_2$ signal with depth in the Dome Fuji ice core to show that the permeation coefficients (product of diffusivity and solubility) of $O_2$ as estimated by Ikeda-Fukazawa et al. (2005) are likely too high (by a factor of $\approx$5 at 242 K). If we assume that this difference in permeation coefficients is due to overestimates of diffusivity, we can lower the Ikeda-Fukazawa et al. (2005) value by a factor of 5 to yield $D_{O2,242K,adjusted} = 5.6 \times 10^{-12}$ m$^2$ s$^{-1}$. As we consider the question of whether gas diffusion can be fast enough to explain the observed rapid loss of *in situ* [14]CO from ice grains at Summit, we use this diffusivity estimate (lowest of the set of values discussed above) to be conservative. The characteristic time scale for diffusion can be estimated as $t = L^2/D$, where L is the diffused distance and D is the diffusivity. Taking a typical firn grain radius as 3 mm (Linow et al., 2012), the time scale for diffusion out of the firn grain is then 0.05 yr (18 days), which is sufficiently fast to be consistent with our observations.

If our hypothesis is correct, then the rate of rapid loss of in situ [14]C from the firn grains would be site-dependent, with less rapid loss predicted at colder sites. If we use the temperature dependence of CH$_4$ diffusivity from Ikeda-Fukazawa et al. (2004), then the diffusivity in ice at a very cold ice core site such as Dome C (T = - 54°C) would be $\approx$7 times lower than at Greenland Summit. However, this lower diffusivity would still be sufficient to allow for complete loss of *in situ* [14]C from firn grains on a time scale faster than 1 year. The lack of retention of in situ [14]C in the firn even at cold sites is consistent with findings from prior measurements of [14]CO and [14]CO$_2$ in the Dome C ice core using a dry extraction technique (de Jong et al., 2004).

Regarding the $\approx$0.5% of *in situ* produced [14]CO that is not lost rapidly from the firn matrix, it may be possible that this [14]CO becomes trapped in microbubbles or at dislocations or grain boundaries in the ice lattice. Microbubbles (closed porosity in firn above the lock-in zone) have been observed at ice core sites in both Antarctica (e.g., Siegenthaler et al., 2005) and Greenland (Petrenko et al., 2013; this study – see Table S2) and would be expected to retain [14]CO in the same way as bubbles in deeper ice. Further movement of [14]CO from microbubbles to open porosity would be controlled by the gas permeation coefficient (product of diffusivity and solubility). As solubility of gases in ice is very low (e.g., Patterson and Saltzman, 2021 and references therein), this process is slow. This is different from the situation of [14]CO immediately following *in situ* production, as in this case [14]CO is already in the ice lattice and only diffusivity matters. Dislocations and grain boundaries tend to concentrate impurities (e.g., Stoll et al., 2021 and references therein), and it may be possible that interactions with other impurities help to retain [14]CO molecules. This trapped [14]CO could then be released more slowly (as mentioned above, our best-estimate leakage rate for this slow process is 0.6% yr$^{-1}$) as the microbubbles disintegrate or as dislocations and grain boundaries come in contact with porosity via the relatively slow process of recrystallization.

## 4.3 Production rates from muon mechanisms

### 4.3.1 Approach for constraining production rates from muon mechanisms

In this study, we require the determined muogenic $^{14}$CO production rates to be consistent with both Taylor Glacier (Dyonisius et al., 2023) and Summit measurements. Dyonisius et al. (2023) determined the possible set of $f_{\mu\text{-}}$ -- $f_{\mu f}$ pairs via comparison of predictions from an ice flow model for Taylor Glacier that included *in situ* $^{14}$CO production with Taylor Glacier measurements. We take the set of $f_{\mu\text{-}}$ -- $f_{\mu f}$ pairs from Dyonisius et al. (2023) that represents the 2σ range (3851 pairs) and perform repeated runs of the model, comparing the model $^{14}$CO results with measurements in ice below the LIZ and in firn air in the LIZ. As different sets of $f_{\mu\text{-}}$ -- $f_{\mu f}$ pairs yield optimal solutions for ice and LIZ firn air, we define the "best" and accepted range of $f_{\mu\text{-}}$ -- $f_{\mu f}$ pairs separately for ice and LIZ air. In each case, the reduced $\chi^2$ value is once again used to define the "best" $f_{\mu\text{-}}$ -- $f_{\mu f}$ pair. The accepted sets of $f_{\mu\text{-}}$ -- $f_{\mu f}$ pairs are those that yield model solutions that fall within the 2σ uncertainty bound from the best solution. The 2σ uncertainty bound is defined separately for ice and LIZ firn air and is equal to the mean 2σ measurement uncertainty for samples within the depth range used for the model-data comparison. The final accepted set of $f_{\mu\text{-}}$ -- $f_{\mu f}$ pairs represents the overlap between the sets accepted for ice and for LIZ firn air.

As discussed above, the $^{14}$CO content in bubbly ice below the LIZ and in LIZ firn air is sensitive to the atmospheric [$^{14}$CO] history in addition to muogenic $^{14}$C production rates. We use several possible atmospheric [$^{14}$CO] scenarios (Section 4.3.2) to ensure that we are capturing the full possible range of accepted of $f_{\mu\text{-}}$ -- $f_{\mu f}$ pairs. For each [$^{14}$CO] scenario, we repeat the firn model runs for each $f_{\mu\text{-}}$ -- $f_{\mu f}$ pair from Dyonisius et al. (2023). Finally, the choice of retention and leakage parameters does have a small impact on the $^{14}$CO content in bubbly ice below the LIZ. We therefore also repeat the above sets of firn model runs with $R_1$, $L_1$ and $F_n$ parameters set to either the best-estimate values, the "Min", or the "Max" values in Table 2.

### 4.3.2 Atmospheric [$^{14}$CO] history scenarios

The "baseline" atmospheric [$^{14}$CO] history for Greenland Summit used in this study (Fig. 5, Scenario 1) was developed by Hmiel et al. (2020) and discussed in their Supplementary Information. This history is based on atmospheric $^{14}$C production as indicated by the sunspot group number and was constructed as follows. We used the longest available record of atmospheric [$^{14}$CO] measurements from Manning et al. (2005) (from New Zealand and Antarctica) and the sunspot group number record from Svalgaard and Schatten (2016). For the period of overlap between these two records, we applied a linear regression of the annual sunspot group number and a 12-month moving average of the Southern Hemisphere (SH) [$^{14}$CO] measurements corrected for the secondary [$^{14}$CO] component (~10%) that arises from CH$_4$ and non-CH$_4$ hydrocarbon oxidation and biomass burning (Manning et al., 2005). This yields the Southern Hemisphere (SH) [$^{14}$CO] -- sunspot group number relationship, which is then applied to the earlier part of the sunspot group number record to construct a SH cosmogenic [$^{14}$CO] history. A constant offset of +6.98 molecules cm$^{-3}$ STP is then applied to the SH [$^{14}$CO] history to account for the higher mean annual [$^{14}$CO] at Greenland Summit as compared to the SH as well as for the secondary component of [$^{14}$CO] (e.g., Jockel and Brenninkmeijer, 2002). Lastly, the seasonal cycle in [$^{14}$CO] (not shown in Figure 5) is developed from fitting a harmonic function to the available records of Northern Hemisphere [$^{14}$CO] from high latitude sites of Alert, Ny Alesund, and Barrow (Jöckel and Brenninkmeijer, 2002). The [$^{14}$CO] seasonal cycle amplitude and the constant Greenland - SH offset were

optimized by trialing a range of values in the firn model to minimize the reduced $\chi^2$ metric for the difference between model and firn air [$^{14}$CO] observations in the 0 – 60 m depth range.

While the baseline atmospheric [$^{14}$CO] history at Summit described above takes into account variations in atmospheric $^{14}$C production rate, atmospheric [$^{14}$CO] is also sensitive to variations in hydroxyl radical concentrations ([OH]), stratosphere-to-troposphere transport (STT), and to a lesser extent, changes in secondary $^{14}$CO sources such

as CH$_4$ and VOC oxidation and biomass burning (e.g., Jöckel and Brenninkmeijer, 2002). Variability in global [OH] is relatively well constrained via observations of methyl chloroform back to the mid-1990s (e.g., Montzka et al., 2011; no large changes), but more uncertain further back in time. With regard to STT, we investigated the sensitivity of [$^{14}$CO] over Greenland to changes in the Brewer-Dobson circulation using a preliminary implementation of $^{14}$CO in the GEOS-Chem chemical transport model. A preindustrial run was compared with a 4xCO$_2$ run in GEOS-Chem (the

Brewer Dobson circulation is expected to intensify with warming), and no significant change in [$^{14}$CO] over Greenland was found.

However, to ensure that plausible $f_{\mu^-}$ -- $f_{\mu f}$ pairs were not excluded, we still allowed for relatively large (up to 30%) variability in [$^{14}$CO] as compared to the baseline scenario, to account for possibilities such as northern hemisphere [OH] variations prior to the 1990s in response to large changes in emissions of reactive species, increasing

uncertainty in the atmospheric $^{14}$C production rates further back in time and possible changes in the frequency of stratospheric air intrusions. The [$^{14}$CO] atmospheric history scenarios we trialed differed as follows from the "baseline" scenario:

1) Baseline scenario
2) Constant [$^{14}$CO] at all times at 18.5 molecules / cm$^3$ STP
3) +15% change from baseline at all times
4) -15% change from baseline at all times
5) -30% change from baseline at all times
6) A linear increase in [$^{14}$CO] back in time from baseline [$^{14}$CO] in 2013, reaching +15% at 1980, then a linear decrease further back in time, reaching -15% at 1850 as compared to baseline; -15% from baseline for ages
older than 1850
7) Same temporal structure as 6), except variations of +30% in 1980 and -30% at 1850 and older
8) Same temporal structure as 6), except variations of +15% in 1980 and -30% at 1850 and older

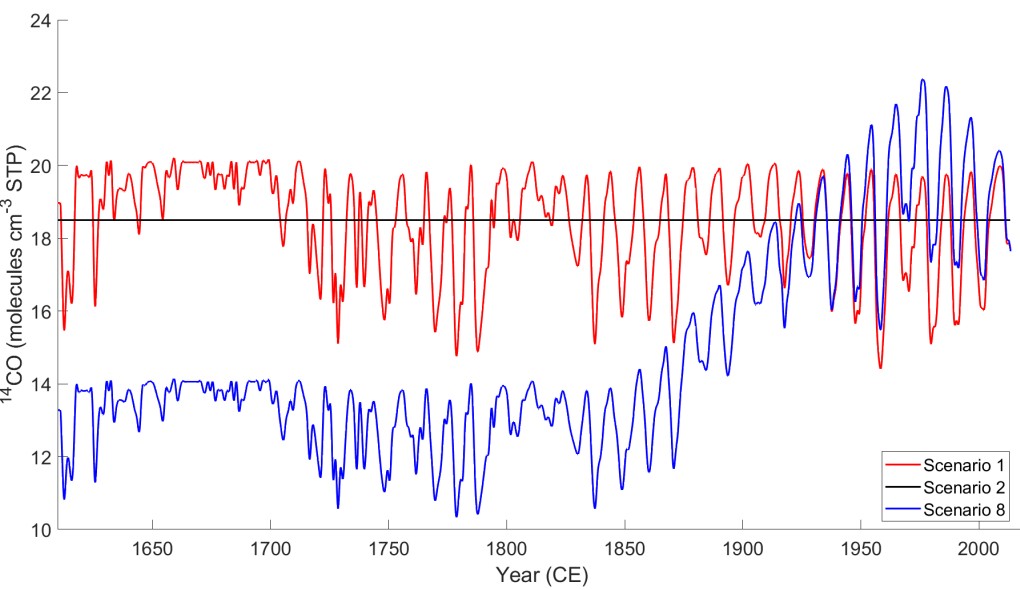

**Figure 5: Paleoatmospheric [$^{14}$CO] scenarios.** Visualization of some of the paleoatmospheric $^{14}$CO history scenarios used in constraining muogenic $^{14}$CO production tuning factors $f_{\mu^-}$ and $f_{\mu f}$. Scenario names in the figure caption correspond to the list in Section 4.3.2.

### 4.3.3 Results for muogenic $^{14}$CO production rate tuning factors $f_{\mu^-}$ and $f_{\mu f}$

Figure 6 shows the results for the trialed combination of atmospheric [$^{14}$CO] history and $R_1$, $L_1$ and $F_n$ parameters that yields the largest accepted set of $f_{\mu^-}$ -- $f_{\mu f}$ pairs. This set, which yields solutions that give a good fit to the measurements in both bubbly ice and LIZ firn air, is bounded by the blue contour in Figure 6c. Table 3 compares the $2\sigma$ ranges for $f_{\mu^-}$ and $f_{\mu f}$ values between this study and Dyonisius et al. (2023). As can be seen, the Summit results are consistent with most of the $f_{\mu^-}$ range from the Taylor Glacier (TG) study, but only with approximately the lower half of the $f_{\mu f}$ range. This is not surprising – the Taylor Glacier results for $f_{\mu f}$ had larger relative uncertainties, due mainly to uncertainties in the depth of ice transport in the glacier over the last few thousand years. Our results are consistent with the relatively shallower Taylor Glacier flowlines considered in Dyonisius et al. (2023).

We note that the atmospheric [$^{14}$CO] history scenarios that result in the best model fit to both the bubbly ice and LIZ firn air measurements all contain significant temporal variability, with [$^{14}$CO] around 1980 that is higher than predicted from estimated atmospheric $^{14}$CO production rates and [$^{14}$CO] prior to ≈1920 that is lower than predicted (Section 4.3.2 and Figure 5). Such variability, if real, may be indicative of long-term changes in northern hemisphere hydroxyl radical concentrations, and could be investigated more effectively at a site with a very high accumulation rate (e.g., Greenland Southeast Dome; Iizuka et al., 2017) where the *in situ* $^{14}$CO component would be minimized.

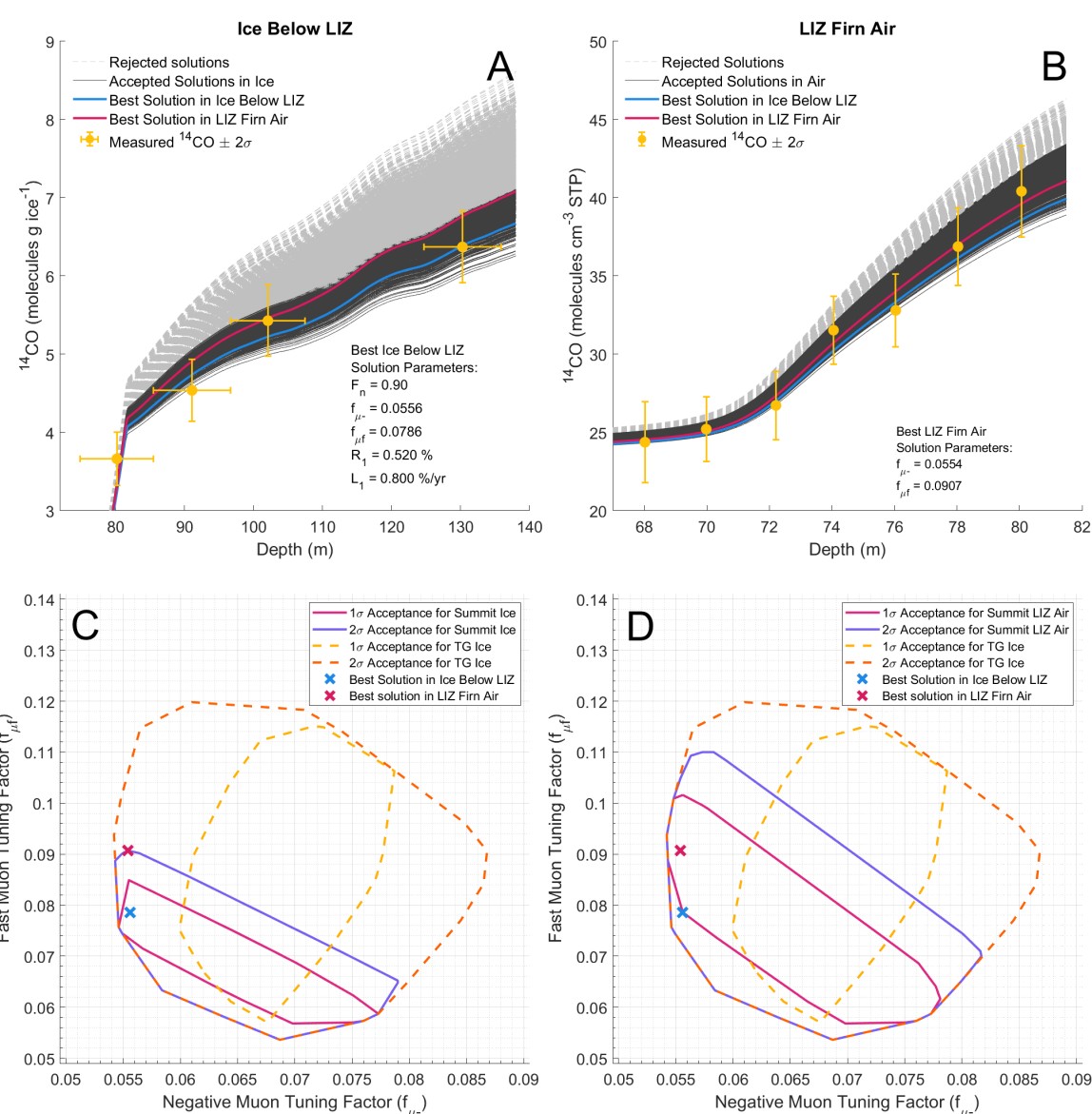

**Figure 6. Constraints on $f_{\mu^-}$ and $f_{\mu f}$ values.** Results from repeated runs of the firn model with the full set of $f_{\mu^-}$ -- $f_{\mu f}$ pairs from Dyonisius et al. (2023). In the shown model runs, atmospheric [$^{14}$CO] history scenario 8 (see Section 4.3.2) was used, along with the set of $F_n$, $R_1$ and $L_1$ values that yield minimum $^{14}$CO content in ice below the LIZ (Table 2). Panel A shows the model run results for $^{14}$CO content in ice below the LIZ, and panel B shows the results for LIZ firn air. Panels C (ice) and D (LIZ firn air) show the corresponding contours of accepted sets of $f_{\mu^-}$ - $f_{\mu f}$ values.

| | Dyonisius et al., 2023 | This work |
|---|---|---|
| $f_{\mu-}$ 2σ range | 0.0542 – 0.0868 | 0.0543 – 0.0790 |
| $f_{\mu f}$ 2σ range | 0.0536 – 0.1198 | 0.0536 – 0.0907 |

**Table 3. Comparison of $f_\mu$ ranges from Dyonisius et al. (2023) and this work.**

### 4.3.4 Discussion of implications for muogenic $^{14}$C production rates

Because Dyonisius et al. (2023) observed a constant *in situ* $^{14}$CO / $^{14}$CO$_2$ production ratio, our results also indirectly confirm their finding that muogenic $^{14}$C production rates for $^{14}$C-bearing gases in ice appear to be much lower than total $^{14}$C production rates predicted from studies in quartz. Dyonisius et al. (2023) considered this discrepancy and were able to rule out the following as possible explanations: 1) analytical issues with ice core $^{14}$C measurements (good agreement between independent methods); 2) much higher historical ice ablation rates as compared to modern (not supported by paleoclimate data; also for Greenland Summit the ice chronology is very well determined so accumulation rate is not in question); 3) $f_{\mu f}$ is zero and $f_{\mu-}$ is much higher (this would be consistent with $^{14}$C results from a rock core, but is not consistent with ice core data); 4) the unmeasured organic fraction of in situ $^{14}$C in ice explains the difference (organic fraction is <25%, so too small); 5) the value used for α (0.75; see Equation 5) is too small (using a upper-end value for α does not change the results appreciably).

If there was a process that destroyed $^{14}$CO to a significant extent in ice at Greenland Summit and other previously sampled locations, biasing the ice core measurements low, this could help to reconcile the observed muogenic $^{14}$C production rate difference between ice and quartz. Such a chemical or biological process would very likely be temperature and/or impurity dependent, and therefore site dependent. There are two previously published ice core $^{14}$CO data sets using different methodology from two different locations in Antarctica: Scharffenbergbotnen (van der Kemp et al., 2002) and Taylor Glacier (Dyonisius et al., 2023) and these agree with each other with regard to production rates within uncertainties; as do Greenland Summit and Taylor Glacier. An additional new $^{14}$CO data set from Law Dome, Antarctica is also consistent with the muogenic production rates determined here (Petrenko et al., 2023). Further, while CO in Greenland ice cores is known to be affected by *in situ* production, CO in Antarctic ice cores appears to be generally well preserved and there is no evidence for CO destruction in either Greenland or Antarctic ice cores (e.g., Fain et al., 2022; Fain et al., 2023; Haan et al., 1998). This process therefore seems highly unlikely to explain the ice – quartz disagreement.

Partitioning of "hot" *in situ* $^{14}$C atoms among different species in the polar ice cores is unlikely to be site-dependent because impurities are present in Greenland and Antarctic ice cores at trace levels (order of part per million or lower), so the probability of a "hot" $^{14}$C atom bonding with atoms other than O and H is very low. The agreement among different ice core sites with regard to $^{14}$CO production rates suggests constant partitioning of muogenic *in situ* $^{14}$C. Variations in "hot" $^{14}$C partitioning in ice therefore also seem unlikely to explain the ice – quartz disagreement.

It may in principle be possible that gas-phase ice core $^{14}$C measurements ($^{14}$CO + $^{14}$CO$_2$ + $^{14}$CH$_4$) are missing the majority of *in situ* $^{14}$C if the majority of this $^{14}$C is present as non-gas species. While the organic fraction of $^{14}$C mentioned above seems too small to be the main explanation, it may be possible that other forms of inorganic carbon (HCO$_3^-$, CO$_3^{2-}$) play a role.

**5 Conclusions**

This study has provided the most comprehensive characterization to date of the production, movement and retention of *in situ* cosmogenic $^{14}$CO in the firn layer of an accumulation site. Our results conclusively show that only a very small fraction (< 0.6%) of total *in situ* cosmogenic $^{14}$CO that is produced in the ice grains initially stays in the ice grains. Measurements in the firn matrix indicate that the vast majority of *in situ* $^{14}$CO is lost very rapidly to the open porosity in the firn. Gas diffusion through ice appears to be the most likely process responsible for this rapid loss. Our firn matrix measurements also conclusively demonstrate that there is a second, much slower process for loss of *in situ* $^{14}$CO from the ice grains that occurs on timescales that could be consistent with the process of firn recrystallization. More measurements from different ice core sites would provide further insight into $^{14}$C loss and retention processes and would allow for explicit inclusion of these physical processes into future models of *in situ* $^{14}$C in the firn.

While the uncertain atmospheric [$^{14}$CO] history over Greenland translates into increased uncertainties in the determination of muogenic $^{14}$CO production rates in ice, we were nevertheless able to further narrow the possible range of these production rates. These findings help to lay the groundwork for ice core studies of past atmospheric [$^{14}$CO], past cosmic ray flux, as well as for ice core studies that use other $^{14}$C-containing gases for applications such as examining the past atmospheric methane budget (using $^{14}$CH$_4$) and the past carbon cycle (using $^{14}$CO$_2$). Finally, our results continue to highlight the apparent discrepancy in muogenic $^{14}$C production rates determined from studies of gases in ice versus those determined from studies in quartz. No clear explanation for this discrepancy currently exists, although it is possible that studies in ice to date have missed some of the species that *in situ* $^{14}$C partitions into. Future studies that provide more comprehensive characterization of $^{14}$C in ice cores (including gas species, non-gas organic and inorganic carbon species) as well as laboratory studies involving muon irradiation of ice samples may be helpful in resolving this disagreement. For completeness, future studies could also explicitly consider uncertainties in all muogenic $^{14}$C production parameters, such as muon fluxes and energies.

**Code availability**

Code for the firn model used in this study is available from the authors upon request.

**Data availability**

All the data presented in this study are available in the tables in the Supplement, as well as from the NSF Arctic Data Center: https://arcticdata.io/catalog/view/doi%3A10.18739%2FA2599Z216.

**Supplement link**

The link to the supplement will be included by Copernicus.

**Author Contribution**

VVP designed the study, with input from BH, CB and JPS. BH, VVP, MND, PP, AMS, CB, MP and MD planned and prepared the field campaigns. BH, VVP, MND, PP, CB, AMS, RB, LD, MD, MP, JAM, MK, XF and AA carried out the fieldwork and sample collection. BH, AMS, BY, QH, VVP, RB, JPS, CH, RFW and IV performed or assisted with

sample processing and measurements. BH, VVP, CB, AMS, MND, JPS and LTM analyzed and interpreted the results. VVP and BH wrote the manuscript, with input from all other authors.

**Acknowledgements**
This work was funded by US NSF Award ARC-1203779 (to VVP) and the Packard Fellowship for Science and Engineering (to VVP). We thank the U.S. Ice Drilling Program for support activities through NSF Cooperative Agreement 1836328. We thank Polar Field Services and 109th Airlift Wing of the New York Air National Guard for

providing field logistical support. We thank S. Englund Michel for providing measurements of $\delta^{13}C$ in the dilution gas, J. McConnell for providing age tie-points for the ice cores and J. Patterson and R. Alley for helpful discussions.

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
