# Peer review of "Characterization of *in situ* cosmogenic 14CO production, retention and loss in firn and shallow ice at Summit, Greenland"

_The Cryosphere, 2023_

## Referee Comment (RC2)

**Review of: "Characterization of in situ cosmogenic [14]CO production, retention and loss in firn and shallow ice at Summit, Greenland"**

*by Benjamin Hmiel et al., submitted to The Cryosphere.*

This manuscript targets understanding and constraint of the production, movement and retention of in-situ cosmogenic [14]C in ice. This is done based on the analysis of [14]CO at different depths in the firn, both, in gas from the porous, open firn space (firn air) and the gas trapped in extracted firn/ice samples (firn matrix and bubbly ice below the firn zone, respectively). The authors achieved to perform the highly challenging analysis of [14]CO in firn and ice with convincingly high accuracy, which is a fantastic achievement. While the contribution to in-situ [14]CO (and total [14]C) from production by neutrons is relatively well studied and seems reasonably well understood, more recent findings indicated that the signal from production by deep-penetrating muons via the negative muon and fast muon capturing mechanisms is lower than one would expect based on the literature by around a factor of 5 (Dyonisius et al., 2023). This is a relevant difference and can have important implications on the interpretation of results in a variety of research areas where cosmogenic isotope production affecting the background is an issue. The manuscript therefore strongly focuses to further investigate this discrepancy. A modelling framework, essentially combing a previously established [14]C production model (Balco et al., 2008; adapted for a firn/ice matrix) with the firn gas transport model by Buizert et al. (2012) and a new box model to consider retention (or leakage, respectively) of the evolving [14]CO partition/fraction to the firn gas was accordingly developed. For the processes and conditions assumed in this model approach and compiled up-to-date atmospheric histories used for input, a close match between model results and the paleo-observational data was achieved but noteworthy, includes a variety of free (tunable) parameters (factors) associated with the various, archive specific (at least partly inter-related) physical and chemical processes and mechanisms involved. The authors show these parameters to be well constrained and thus reasonably close predictions of the expected cosmogenic [14]C in-situ contribution to [14]CO (and likely also [14]CO$_2$) should be achievable, which is certainly valuable and beneficial for future studies in firn and ice.

The paper is very well written, the analytical methods are of highest standard (pushing the boundaries) as are the technical aspect of the modeling. However, while the study confirms the previous findings of lower-than-expected contribution from the negative muon and fast muon capturing mechanisms, it needs to be seen if a revision of the respective production rate estimates is required or if a lack of understanding in many of the complex (and directly related) processes in ice and firn currently remains the more likely explanation. My main concern is linked to this last point (see details below), and I suggest the manuscript to be published after minor reviews.

**Main issues**

General:

The topics covered by the manuscript, from the analytics to the postprocessing of measured data as well as interpretation, are manifold and rather complex. Therefore, and although the authors already did a great job in writing, the manuscript is challenging to read and comprehend. What I

struggled most with, was to keep the overview what model parameters/input and mechanisms are well (or reasonably well) determined based on previous studies and which are introduced factors, required to allow matching of model results with the data (model tuning factors; to just name some examples e.g. R, $f_{u-}$, $f_{uf}$ or factors introduced to account for (additional?) uncertainty like $F_n$). I thus suggest providing an overview table, where the relevant parameters and reconstructed input (e.g. $P_n(0)$, "baseline" atmospheric $[^{14}CO]$) is summarized including the relevant description and some associated information (e.g. uncertainty).

Such an overview would not only facilitate reading, but also be beneficial for the reader to understand if a tight match between model result and data or a well constrained factor is largely the result of an in-depth understanding of processes and mechanisms at play or at least partly the result of a sufficient high number of free parameters allowing for tight model tuning. With "processes and mechanisms at play" I hereby refer to the physical and chemical processes involved and happening in the ice. For the different species, both in the gaseous (e.g. CO, $CO_2$, $CH_4$) and liquid phase (e.g. DOC) the following come to mind: (i) the specific chemical reaction mechanisms and reaction kinetics on ice surfaces/in quasi-liquid-layers, maybe associated with fractionation and considering equilibria in the partitioning of in-situ $^{14}C$ into different species (e.g. CO, $CO_2$, $CH_4$, DOC) potentially also temperature dependent, (ii) the diffusion of gases in ice, which is certainly different from the diffusion of DOC (and also its releases into the porous open space of the firn), (iii) snow and firn metamosrphism (i.e. recristalisation), (iv) the potential effect of impurities in ice on $^{14}C$ production rates, (v) the  gas transport in the firn and firn ventilation, etc.. Some of these points are more or less thoroughly addressed and discussed in the current manuscript while others are not mentioned. For this reason, I suggest that a revised version of the manuscript aims to better clarify and distinguish the level of process/mechanism understanding and the likelihood that a lack thereof might (or might not) explain the observed discrepancy, e.g. between the observed $^{14}CO$ in ice and the one expected based on previous determinations of $^{14}C$ production rates from studies in quartz.

The authors should note that for in-situ cosmogenic $^{14}C$ analysis in quartz the analytical procedures and techniques are very well established and a very large number of studies exist, the use of reference standard reference materials for inter-laboratory comparison is common practice (e.g. Lupker et al., 2019; Nichols et al., 2022). Generally, the analysis in quartz is likely a more direct measurement than in ice, because potential processes in the archive (i.e. quartz) are assumingly less and easier to understand compared to the many (not fully understood factors) in firn and ice discussed in the manuscript and supplemented in the paragraph above. Therefore, statements like in line 614 ff. "…our results also indirectly confirm … that muogenic $^{14}C$ production rates in ice are several times lower than what would be predicted from studies in quartz – a puzzle that currently lacks a good explanation." should be put a bit more into context (also see line 42 in the abstract or L 647 ff in the conclusions). Obvious to me, the by far most likely explanation seems to be that the processes in firn and ice are still not fully understood yet.

Detailed:

L 54 ff. "The in situ produced $^{14}C$ mainly forms $^{14}CO_2$ and $^{14}CO$, with a smaller fraction forming $^{14}CH_4$ and possibly other simple organics such as formaldehyde (Dyonisius et al., 2023; Hoffman, 2016; van der Kemp et al., 2002; Fang et al., 2021)." Of the total number of in-situ produced $^{14}C$ atoms per gram ice, Hoffmann (2016) found a fraction of 11-25 % incorporated into the DOC fraction when

performing a neutron irradiation experiment on Alpine ice core samples. The incorporation of cosmogenic in-situ $^{14}C$ into DOC has later been supported by measurements in environmental samples (Fang et al., 2021). In Dyonisius et al., 2023 (and obviously all studies before 2022 or at least 2016) this partition has not been considered and is obviously also missing in the reviewed study here (see next point). Unlike $^{14}CH_4$ this does not seem to be a minor fraction and should be considered (maybe needed therefore: DOC concentrations, around 5 $\mu$g C / kg ice for polar ice; Preunkert et al. 2011). As you assume for CO, same partitioning for the n and muon mechanism needs to be similarly assumed here. Important might also be that the cosmogenic produced $^{14}C$ incorporated into DOC is likely to behave differently in the firn/ice than the gaseous species (CO etc) in terms of diffusion and release into the porous firn (and transport therein), basically being fixated after incorporation (removed $^{14}C$ partition in subsequent modeling of firn gas transport/retention/leakage).

L 319 ff. "We use a value of $\Omega^{CO}$= 0.31 for the fraction of total in situ $^{14}C$ in ice that forms $^{14}CO$ (Dyonisius et al., 2023; van der Kemp et al., 2002)." In Dyonisius et al. (2023), the value for $\Omega^{CO}$ seems to be 33.7 % associated with an uncertainty of ±11.4 %. Has an uncertainty for $\Omega^{CO}$ been considered and propagated here? This seems relevant considering the narrow range of $\mu_-$ and $\mu_f$ and the difference compared to earlier values (see Table 2). If not, this should not be too difficult to be introduced for example in a similar way as done for the uncertainty of the n production rate (introduction of an additional, adjustable "uncertainty factor" like $F_n$ in L 326). Also see above regarding the missing partition which is incorporated into DOC.

L 356,357 "…(we use $\alpha$=0.75, consistent with Dyonisius et al., 2023 and Heisinger et al., 2002b),…". Heisinger et al. (2002b) considered uncertainties in the muon energy and flux to be in the range of 10% each. It does not seem you similarly considered and propagated these uncertainties. Aiming for a comprehensive study, propagating all uncertainties will finally be more useful, resulting in the most realistic range estimates. Especially if there still persist a number of unknowns (i.e. a detailed understanding of all process and mechanisms involved). If not considered, the factors or uncertainties should at least be summarized in the discussion (reference to suggested additional table?), providing a possible tie-point for future studies.

L 378 ff. "We introduce these two reservoirs because a preliminary analysis showed that using a single ice grain reservoir does not provide a good fit to the observations." Is there any hypothesis of a possible mechanism/process to justify this partitioning into two reservoirs? What is considered as a good fit (especially if considering all other uncertainties, including the ones mentioned above and some of the points mentioned below), i.e. how bad would the fit be?

Section 4.2, L 481 ff. Is ventilation not part of gas transport/movement in the firn? Since you are using a frin transport model, I do not see why your proposed mechanism would not also (to some extent at the least) include ventilation. Further, snow metamorphism (the most common type of recrystallization of snow and the uppermost firn) is a very fast process (days to weeks); e.g. Pinzer et al., 2012. With depth and higher density of the firn, the recrystallization process will become slower, probably in the order of a few years at the LIZ (e.g. Duval et al., 1995). Thus, would it not be more appropriate to, in the model, combine the processes of diffusion and metamorphism / recrystallization, essentially leading to an "enhanced diffusion" (strong enhancement in the upper firn and much less close/at the LIZ)? Maybe a partitioning of the model into two reservoirs could then be avoided?

**Minor issues**

L 187, equation 1. Definition of $x_{CO}$ seems to be missing in the text.

L 188 "...pMC is the sample or blank $^{14}$C activity in pMC units...". pMC denotes percent modern carbon, a pMC unit does not exist (as the name tells, it is a percentage).

L 191 (& equation 1) "1.1694 × 10$^{-12}$ is the $^{14}$C / ($^{13}$C + $^{12}$C) ratio corresponding to the absolute international $^{14}$C standard activity (Hippe and Lifton, 2014), ..." pMC is defined based on a half life of 5730 (as you mention elsewhere, the half life you used), but Hippe and Lifton, in their reformulation, from which your value of 1.1694 × 10$^{-12}$ results, considered a half-life of 5700. Note that their reformulation is performed for two main reasons: (i) to omit a necessary correction in activity for the decay from 1950 to the year of measurement of the international $^{14}$C standard used for AMS calibration (yielding the number for the activity to be used with 1.1694 × 10$^{-12}$) and (ii) to account for the conventionally introduced $^{13}$C normalization for AMS measurements (to -25 per mil; not to be confused with the additionally performed normalization of the standard to -19 per mil to account for the switch to a new material of the international AMS reference standard; see e.g. https://www.hic.ch.ntu.edu.tw/AMS/A%20guide%20to%20radiocarbon%20units%20and%20calculat ions.pdf). In any case, the reformulation eventually leads to their Eq 22, which in principle corresponds to your Eq 1 (without the terms accounting for CO molecules per volume). Your equation misses one important term (since this is one of the two main reasons to perform the reformulation in the first place), Ab12$_s$, which results from accounting/correcting for $^{13}$C normalization. I suggest checking carefully (maybe using the activity of the reference standard and correcting to the year of measurement might be more appropriate for your data anyhow; not that all that matters much with regards to all other uncertainties, I think...).

L 297 "Figure 1 illustrates total $^{14}$C production rates by each mechanism versus depth at Summit, ...". The term "each mechanism" might not be ideal with regards to the previous sentence where also the mechanism of "loss of this $^{14}$C from the ice grains via leakage into the open porosity (firn air) or closed porosity (air bubbles)." is mentioned. Maybe clearer would be, "Figure 1 illustrates total $^{14}$C production rates by secondary cosmic ray neutrons and muons versus depth at Summit, ...", even though not the most pretty due to repetition.

L 355 "...$\beta(h)$ is a unitless depth dependence factor...". Shouldn't that be mass-depth dependence factor?

**References**

Lupker, M and 7 others (2019) In-situ cosmogenic 14C analysis at ETH Zürich: characterization and performance of a new extraction system. Nuclear Instruments and Methods in Physics Research Section B: Beam Interactions with Materials and Atoms 457(July), 30–36. doi: 10.1016/j.nimb.2019.07.028

Nichols KA. A decade of in situ cosmogenic 14C in Antarctica. Annals of Glaciology. 2022;63(87-89):67-72. doi:10.1017/aog.2023.13

Susanne Preunkert, Michel Legrand, P. Stricker, S. Bulat, Irina Alekhina, et al.. Quantification of Dissolved Organic Carbon at Very Low Levels in Natural Ice Samples by a UV-Induced Oxidation Method. Environmental Science and Technology, 2011, 45 (2), pp.673-678. 10.1021/es1023256.insu-00605044

Pinzer, B. R., Schneebeli, M., and Kaempfer, T. U.: Vapor flux and recrystallization during dry snow metamorphism under a steady temperature gradient as observed by time-lapse micro-tomography, The Cryosphere, 6, 1141–1155, https://doi.org/10.5194/tc-6-1141-2012, 2012.

P. Duval, O. Castelnau. Dynamic Recrystallization of Ice in Polar Ice Sheets. J. Phys. IV,1995, 05 (C3), pp.C3-197-C3-205. 10.1051/jp4:1995317. jpa-00253683

Whang et al., J. Phys. Chem. A, 2023, 127, 2, 517–526, https://doi.org/10.1021/acs.jpca.2c07554

Y. S. Yi, Y. Han, K. D. Kwon, S. K. Lee and S. D. Hur ACS Earth and Space Chemistry 2021 Vol. 5 Issue 11 Pages 3258-3267 DOI: 10.1021/acsearthspacechem.1c00308 https://doi.org/10.1021/acsearthspacechem.1c00308

---

## Author Comment (AC1)

**Response to Anonymous Referee 1**
Referee comments are in regular font
*Our responses are in italics*

Review of the paper « Characterization of in situ cosmogenic 14CO production, retention and loss in firn and shallow ice at Summit, Greenland

The crucial point in this paper is to put additional constrains on the muon induced 14C production rate in ice as function of depth. A recent study (Dyonisius 2023) indicated that this production channel, which is most relevant in depth below the LIZ, seems to be less pronounced than previously predicted in the literature (by a factor of 6 and 4 for negative muon capture and fast muon interaction respectively). These observations were confirmed in the current study. This may have important implication for ice core research, but potentially also for other research areas where cosmogenic isotope production (not only 14C) in the underground becomes relevant. In the conclusions, this aspect could be highlighted more. This holds also for hypothesis regarding the possible underlying reasons for the resulting disagreement, (such as variations in fast muon energy spectra with depth and cross-sectional considerations etc).
*We would argue that the novel / more important contribution of this paper is actually the detailed characterization of in situ $^{14}C$ behavior (retention, loss) in the firn column, which had not been previously done. Our results are indeed consistent with Dyonisius et al (2023) muogenic production rates for $^{14}CO$. However, this study only considers $^{14}CO$, which at most amounts to 1/3 of total in situ $^{14}C$ in ice. We therefore do not think that this manuscript is the right place for an in-depth discussion of the ice vs quartz discrepancy in muogenic $^{14}C$ production rates. Referee 2 had some similar suggestions (with more detail); please see our response to Referee 2 also. To address this comment and the associated comments from Referee 2, we will nevertheless add some more discussion of this issue in the revised manuscript.*

The paper is very well written and technically sound (with some minor issues as listed below). The 14C extraction and analytical methods, which follow well established procedures described in the literature are well documented. Data interpretation was done in the frame of existing production- (Balco 2008) and ice transport models (Buizert 2012). This excellent contribution is basically ready to be published as it is.
*We thank the referee for their positive and constructive review.*

Some uncertainties persist in the parametrization of the partitioning of 14CO in the ice reservoirs. The introduction of a two-domain approach, involving rapid and slower release rates/reservoirs, may seem somewhat arbitrary. While this approach might align well with the data (as expected with the inclusion of extra model parameters), there remains a need for some additional physical justification in the text, although some hypotheses are mentioned in L 526 ff. The small resulting retention of ~0.5% makes me wonder how "worst" the model fit would be without consideration of R1?
*Physical justification / mechanism for the main, rapidly-leaking $^{14}CO$ reservoir / process in the model ($R_0$, $L_0$) is hypothesized to be gas diffusion through ice and is presented in detail in the manuscript in section 4.2, starting on line 493. Regarding the physical justification / mechanism*

*for the small, slowly-leaking $^{14}CO$ reservoir / process in the model ($R_1$, $L_1$), we can add some further details, for example highlighting the evidence for microbubbles (closed porosity) in the shallower part of the firn, both from our results and prior studies.*
*While the fraction of in situ cosmogenic $^{14}CO$ that is initially retained in the firn matrix (≈0.5%) is small, it is not possible to explain the results without this smaller slow-leaking reservoir ($R_1$). If R1 is ignored (R1 = 0), then there is no retention of 14CO in the firn matrix, and all the model curves on Fig. 4A would have 14CO = 0 in the depth range from 0 to ≈60 m. So there is no way that the model can fit the data without considering R1.*

The leakage coefficient L0 was fixed at 1 yr-1 (corresponding to a half-loss time of 0.7 years). Where is this value coming from?
*We thank the reviewer for pointing this out. This is an error in the manuscript description of this part of the model (though not in the model calculations). The primary (larger, fast-leaking) ice grain reservoir is defined in the model in such a way that all of the $^{14}C$ in this reservoir is lost to the porosity during each time step (time step = 0.5 yrs). We will correct the error and clarify this in the manuscript.*

What exactly is meant with "indicating slow leakage of 14CO from the ice grain (L 421 p 13) in relation to the concentration decrease with depth? Please clarify.
*As firn layers move downwards in the firn via the processes of continued snow accumulation at the surface / firn densification, the layers would carry with them the ice grain $^{14}CO$ content they acquired above. Further, $^{14}CO$ production continues (by muons) at intermediate firn depths (20 – 60 m). Therefore, the fact that $^{14}CO$ content in the ice grains decreases rather than increases with increasing depth between 20 and 60m indicates slow $^{14}CO$ leakage (loss) out of the ice grains. $^{14}C$ radioactive decay is far too slow to explain this, as the ice layers traverse the entire firn column at Greenland Summit in only ≈200 years. We will clarify this further in the revised manuscript.*

Minor issues
P14 L455ff: there is something wrong with the grid search interval for Fn. I guess the step size should be 0.01 instead of 0.05
*We thank the referee for catching this typo -- this should indeed state 0.01; we will correct this in the revised manuscript*

Page 16, L515ff: Also, here is something wrong. It seems that the diffusion time was calculated with a grain radius of 3mm. Either this is a typo or the resulting diffusion time for 0.3mm should be ~4.5 h instead of 18 days (what emphasizes the assumption of nearly complete 14C loss on timescales of 1 year even more).
*We thank the referee for catching this typo as well -- we indeed calculated the diffusion time using a radius of 3 mm; this will be corrected in the revised text*

---

## Author Response (AR2)

Referee comments are in regular font
*Our responses are in italics*
*NOTE: due to a software bug in Microsoft Word, the line numbering is a bit different between the "clean" and "Track Changes" versions of the revised manuscript. The line numbers we reference below correspond to the "track changes" version*
*NOTE added 5 June 2024: Further technical corrections requested by the editor are addressed at the end of this document*

**Response to Anonymous Referee 1**
Review of the paper « Characterization of in situ cosmogenic 14CO production, retention and loss in firn and shallow ice at Summit, Greenland

The crucial point in this paper is to put additional constrains on the muon induced 14C production rate in ice as function of depth. A recent study (Dyonisius 2023) indicated that this production channel, which is most relevant in depth below the LIZ, seems to be less pronounced than previously predicted in the literature (by a factor of 6 and 4 for negative muon capture and fast muon interaction respectively). These observations were confirmed in the current study. This may have important implication for ice core research, but potentially also for other research areas where cosmogenic isotope production (not only 14C) in the underground becomes relevant. In the conclusions, this aspect could be highlighted more. This holds also for hypothesis regarding the possible underlying reasons for the resulting disagreement, (such as variations in fast muon energy spectra with depth and cross-sectional considerations etc).
*We would argue that the novel / more important contribution of this paper is actually the detailed characterization of in situ $^{14}C$ behavior (retention, loss) in the firn column, which had not been previously done. Our results are indeed consistent with Dyonisius et al (2023) muogenic production rates for $^{14}CO$. However, this study only considers $^{14}CO$, which at most amounts to 1/3 of total in situ $^{14}C$ in ice. We therefore do not think that this manuscript is the best place for an in-depth discussion of the ice vs quartz discrepancy in muogenic $^{14}C$ production rates. However, as both referees highlighted this point, we have made some edits to highlight that this study's findings are focused on 14CO and added a new section discussing the apparent production rate discrepancy between ice and quartz. Please see our detailed responses to Referee 2.*

The paper is very well written and technically sound (with some minor issues as listed below). The 14C extraction and analytical methods, which follow well established procedures described in the literature are well documented. Data interpretation was done in the frame of existing production- (Balco 2008) and ice transport models (Buizert 2012). This excellent contribution is basically ready to be published as it is.
*We thank the referee for their positive and constructive review.*

Some uncertainties persist in the parametrization of the partitioning of 14CO in the ice reservoirs. The introduction of a two-domain approach, involving rapid and slower release rates/reservoirs, may seem somewhat arbitrary. While this approach might align well with the data (as expected with the inclusion of extra model parameters), there remains a need for some

additional physical justification in the text, although some hypotheses are mentioned in L 526 ff. The small resulting retention of ~0.5% makes me wonder how "worst" the model fit would be without consideration of R1?

*Physical justification / mechanism for the main, rapidly-leaking $^{14}CO$ reservoir / process in the model ($R_0$, $L_0$) is hypothesized to be gas diffusion through ice and was already presented in detail in the original version of the manuscript in section 4.2. Regarding the physical justification / mechanism for the small, slowly-leaking $^{14}CO$ reservoir / process in the model ($R_1$, $L_1$), which we hypothesize to be related to microbubbles and possibly also interactions with impurities at grain boundaries and dislocations, we added some brief discussion, starting on lines 425 and 596 in the revised manuscript.*

*While the fraction of in situ cosmogenic $^{14}CO$ that is initially retained in the firn matrix ($\approx$0.5%) is small, it is not possible to explain the results without this smaller slow-leaking reservoir ($R_1$). If R1 is ignored (R1 = 0), then there is no retention of 14CO in the firn matrix, and all the model curves on Fig. 4A would have 14CO = 0 in the depth range from 0 to $\approx$60 m. So there is no way that the model can fit the data without considering R1.*

The leakage coefficient L0 was fixed at 1 yr-1 (corresponding to a half-loss time of 0.7 years). Where is this value coming from?

*We thank the reviewer for pointing this out. This is an error in the manuscript description of this part of the model (though not in the model calculations). The primary (larger, fast-leaking) ice grain reservoir is defined in the model in such a way that all of the $^{14}C$ in this reservoir is lost to the porosity during each time step (time step = 0.5 yrs). We have corrected and clarified this in Section 3.2.3 of the revised manuscript*

What exactly is meant with "indicating slow leakage of 14CO from the ice grain (L 421 p 13) in relation to the concentration decrease with depth? Please clarify.

*As firn layers move downwards in the firn via the processes of continued snow accumulation at the surface / firn densification, the layers would carry with them the ice grain $^{14}CO$ content they acquired above. Further, $^{14}CO$ production continues (by muons) at intermediate firn depths (20 – 60 m). Therefore, the fact that $^{14}CO$ content in the ice grains decreases rather than increases with increasing depth between 20 and 60m indicates slow $^{14}CO$ leakage (loss) out of the ice grains. $^{14}C$ radioactive decay is far too slow to explain this, as the ice layers traverse the entire firn column at Greenland Summit in only $\approx$200 years. We have added this clarification in the revised manuscript, starting on line 477.*

Minor issues
P14 L455ff: there is something wrong with the grid search interval for Fn. I guess the step size should be 0.01 instead of 0.05
*We thank the referee for catching this typo -- this should indeed state 0.01; we corrected this in the revised manuscript*

Page 16, L515ff: Also, here is something wrong. It seems that the diffusion time was calculated with a grain radius of 3mm. Either this is a typo or the resulting diffusion time for 0.3mm should

be ~4.5 h instead of 18 days (what emphasizes the assumption of nearly complete 14C loss on timescales of 1 year even more).

*We thank the referee for catching this typo as well -- we indeed calculated the diffusion time using a radius of 3 mm; this has been corrected in the revised text*

Characterization of in situ cosmogenic 14CO production, retention and loss in firn and shallow ice at Summit, Greenland
Benjamin Hmiel
The Cryosphere

**Response to Anonymous Referee 2**

**Review of: "Characterization of in situ cosmogenic 14CO production, retention and loss in firn and shallow ice at Summit, Greenland"**
*by Benjamin Hmiel et al., submitted to The Cryosphere.*
This manuscript targets understanding and constraint of the production, movement and retention of in-situ cosmogenic 14C in ice. This is done based on the analysis of 14CO at different depths in the firn, both, in gas from the porous, open firn space (firn air) and the gas trapped in extracted firn/ice samples (firn matrix and bubbly ice below the firn zone, respectively). The authors achieved to perform the highly challenging analysis of 14CO in firn and ice with convincingly high accuracy, which is a fantastic achievement. While the contribution to in-situ 14CO (and total 14C) from production by neutrons is relatively well studied and seems reasonably well understood, more recent findings indicated that the signal from production by deep-penetrating muons via the negative muon and fast muon capturing mechanisms is lower than one would expect based on the literature by around a factor of 5 (Dyonisius et al., 2023). This is a relevant difference and can have important implications on the interpretation of results in a variety of research areas where cosmogenic isotope production affecting the background is an issue. The manuscript therefore strongly focuses to further investigate this discrepancy. A modelling framework, essentially combing a previously established 14C production model (Balco et al., 2008; adapted for a firn/ice matrix) with the firn gas transport model by Buizert et al. (2012) and a new box model to consider retention (or leakage, respectively) of the evolving 14CO partition/fraction to the firn gas was accordingly developed. For the processes and conditions assumed in this model approach and compiled up-to-date atmospheric histories used for input, a close match between model results and the paleo-observational data was achieved but noteworthy, includes a variety of free (tunable) parameters (factors) associated with the various, archive specific (at least partly inter-related) physical and chemical processes and mechanisms involved. The authors show these parameters to be well constrained and thus reasonably close predictions of the expected cosmogenic 14C in-situ contribution to 14CO (and likely also 14CO2) should be achievable, which is certainly valuable and beneficial for future studies in firn and ice.
The paper is very well written, the analytical methods are of highest standard (pushing the boundaries) as are the technical aspect of the modeling. However, while the study confirms the previous findings of lower-than-expected contribution from the negative muon and fast muon capturing mechanisms, it needs to be seen if a revision of the respective production rate estimates is required or if a lack of understanding in many of the complex (and directly related) processes in ice and firn currently remains the more likely explanation. My main concern is linked to this last point (see details below), and I suggest the manuscript to be published after minor reviews.
*We thank the referee for their very in-depth review, and address the points below.*

**Main issues**
General:

The topics covered by the manuscript, from the analytics to the postprocessing of measured data as well as interpretation, are manifold and rather complex. Therefore, and although the authors already did a great job in writing, the manuscript is challenging to read and comprehend. What I struggled most with, was to keep the overview what model parameters/input and mechanisms are well (or reasonably well) determined based on previous studies and which are introduced factors, required to allow matching of model results with the data (model tuning factors; to just name some examples e.g. R, fu-, fuf or factors introduced to account for (additional?) uncertainty like Fn). I thus suggest providing an overview table, where the relevant parameters and reconstructed input (e.g. Pn(0), "baseline" atmospheric [14CO]) is summarized including the relevant description and some associated information (e.g. uncertainty).

*We have included such an overview table (now Table 1) in the revised manuscript to assist readers with keeping track of the model parameters.*

Such an overview would not only facilitate reading, but also be beneficial for the reader to understand if a tight match between model result and data or a well constrained factor is largely the result of an in-depth understanding of processes and mechanisms at play or at least partly the result of a sufficient high number of free parameters allowing for tight model tuning.

*To clarify, our main goal in the model-data comparison is not to arrive at a single set of tunable parameters that yields the best match to the data, but rather to determine the possible range of values for each of the tunable parameters.*

*There are indeed several adjustable parameters in the model ($R_1$, $L_1$, $F_n$, $f_{\mu-}$, $f_{\mu f}$, atmospheric $^{14}CO$ history). However, we are only trying to determine the possible ranges of $R_1$, $L_1$, $f_{\mu-}$, $f_{\mu f}$, and not all at once. $F_n$ (0.9 – 1.1 value range) is simply a scaling factor that represents the uncertainty range for $^{14}C$ production rate by neutrons from prior studies (see Section 3.2.1 as well as line 452 in the original manuscript). The range of atmospheric 14CO histories used in the model (Section 4.3.2) is intended to be conservative (as broad as possible) considering possible variations in the main $^{14}CO$ sink (OH radicals) and stratosphere to troposphere transport. The broad range of 14CO histories chosen ensures that the range of accepted $f_{\mu f}$ and $f_{\mu-}$ values is also conservative.*

*The $R_1 – L_1$ parameter ranges are determined based on the firn matrix $^{14}CO$ profile (Figure 4; lines 431 – 439 already explained this in the original manuscript) and are insensitive to the choice of $f_{\mu-}$ and $f_{\mu f}$, as the original manuscript already also already explains on lines 453 – 457. So here we are only determining two parameters at once, and provide feasible ranges for both.*

*The $f_{\mu-}$ -- $f_{\mu f}$ parameter ranges are determined based on $^{14}CO$ measurements in lock-in zone (LIZ) firn air and in ice below the LIZ (already stated on lines 440 – 442 in the original manuscript). As Section 4.3.1 in the manuscript explains, in the model we trial all $f_{\mu-}$ -- $f_{\mu f}$ parameter pairs from Dyonisius et al. (2023), together with several options of atmospheric $^{14}CO$ histories as well as combinations of $F_n$, $R_1$ and $L_1$ that yield the full possible range of resulting $^{14}CO$ (from maximum to minimum). Trialing a wide range of scenarios for atmospheric $^{14}CO$ history and the full*

*possible range of values for $F_n$, $R_1$ and $L_1$ (as determined in Section 4.2 of the manuscript) allows for a conservative (broad) range of $f_{\mu-}$ -- $f_{\mu f}$ values to be accepted.*

*We added a further clarification in the manuscript that we are only ever constraining two tunable parameters at a time; this starts on line 458 in the revised manuscript.*

With "processes and mechanisms at play" I hereby refer to the physical and chemical processes involved and happening in the ice. For the different species, both in the gaseous (e.g. CO, CO2, CH4) and liquid phase (e.g. DOC) the following come to mind: (i) the specific chemical reaction mechanisms and reaction kinetics on ice surfaces/in quasi-liquid-layers, maybe associated with fractionation and considering equilibria in the partitioning of in-situ 14C into different species (e.g. CO, CO2, CH4, DOC) potentially also temperature dependent, (ii) the diffusion of gases in ice, which is certainly different from the diffusion of DOC (and also its releases into the porous open space of the firn), (iii) snow and firn metamosrphism (i.e. recristalisation), (iv) the potential effect of impurities in ice on 14C production rates, (v) the gas transport in the firn and firn ventilation, etc.. Some of these points are more or less thoroughly addressed and discussed in the current manuscript while others are not mentioned. For this reason, I suggest that a revised version of the manuscript aims to better clarify and distinguish the level of process/mechanism understanding and the likelihood that a lack thereof might (or might not) explain the observed discrepancy, e.g. between the observed 14CO in ice and the one expected based on previous determinations of 14C production rates from studies in quartz.

The authors should note that for in-situ cosmogenic 14C analysis in quartz the analytical procedures and techniques are very well established and a very large number of studies exist, the use of reference standard reference materials for inter-laboratory comparison is common practice (e.g. Lupker et al., 2019; Nichols et al., 2022). Generally, the analysis in quartz is likely a more direct measurement than in ice, because potential processes in the archive (i.e. quartz) are assumingly less and easier to understand compared to the many (not fully understood factors) in firn and ice discussed in the manuscript and supplemented in the paragraph above. Therefore, statements like in line 614 ff. "...our results also indirectly confirm ... that muogenic 14C production rates in ice are several times lower than what would be predicted from studies in quartz – a puzzle that currently lacks a good explanation." should be put a bit more into context (also see line 42 in the abstract or L 647 ff in the conclusions). Obvious to me, the by far most likely explanation seems to be that the processes in firn and ice are still not fully understood yet.

*For this manuscript, the only in situ cosmogenic $^{14}C$ species that we have characterized is $^{14}CO$. Prior studies (van der Kemp et al., Tellus B, 2002, Dyonisius et al., Cryosphere, 2023) indicated that approximately twice as much of the in situ $^{14}C$ in ice forms $^{14}CO_2$ as compared to $^{14}CO$. $^{14}CO_2$ in Greenland Summit ice and firn is strongly dominated by the trapped atmospheric component, unfortunately precluding precise determination of the in situ cosmogenic $^{14}CO_2$ component. Since we are at most examining $\approx 1/3$ of the total in situ cosmogenic $^{14}C$ in Summit ice and firn, we intentionally did not focus too strongly on the disagreement between our findings for muogenic $^{14}C$ production rates in ice and the prior estimates in quartz. Dyonisius et al 2023, in contrast, measured $^{14}CO_2$, $^{14}CO$ and $^{14}CH_4$ and were therefore better positioned to make the comparison to the quartz estimates.*

*However, as both referees highlighted the importance of discussing the ice vs quartz discrepancy in muogenic 14C production rates, we have now included a new section in the text (4.3.4) to address this (as well as a couple of sentences in conclusions). The new section mentions the possibility that carbon species previously not measured by in situ 14C studies in ice (such as bicarbonate and carbonate ions) may be important in explaining the discrepancy. As, again, the study's main focus is 14CO only, we prefer to have the abstract focus on 14CO.*

*Regarding the specific mechanisms the referee mentions above as possible explanations for why muogenic $^{14}C$ production rates in ice may appear lower than estimates in quartz:*

    *(i)    In situ chemical reactions effect on $^{14}CO$ would indeed be expected to be temperature and possibly impurity-dependent. There are now 4 studies from 4 different locations (Dyonisius et al., 2023 – Taylor Glacier, Antarctica; van der Kemp et al., 2023 – Scharffenbergbotnen, Antarctica; this study – Greenland Summit; new results from Law Dome, Antarctica) that have different ice temperatures and impurity loadings. Muogenic $^{14}CO$ production rates for all of these studies agree within uncertainties, arguing against such reactions destroying in situ $^{14}CO$ to a significant extent. Partitioning of $^{14}C$ among different species (e.g., $^{14}CO$, $^{14}CO_2$) is due to "hot" (high energy) atom reactions and as such would not be expected to be affected by ice temperature or differences in concentration of trace impurities. We added some brief discussion of this in the revised manuscript (new Section 4.3.4)*

    *(ii)    Diffusion of gases through the ice lattice has already been fully considered (see discussion in Section 4.2)*

    *(iii)    Firn metamorphism likely affects the release of $^{14}CO$ from ice grains into firn porosity, but it would not be expected to destroy $^{14}CO$. Further, $^{14}CO$ released into porosity can be detected by our firn air measurements. So this cannot be the mechanism that results in lower total $^{14}C$ content.*

    *(iv)    This is related to our response in (i) above. Impurities in ice in interior Greenland and Antarctica are generally present at very low (part per million) levels. They would therefore not significantly affect the abundance of $^{16}O$ target nuclei for muogenic $^{14}C$ production, nor would they be expected to affect the partitioning of in situ $^{14}C$ between species, as the "hot" $^{14}C$ atoms will encounter mainly O and H atoms. We added some brief discussion of this in the revised manuscript (new Section 4.3.4)*

    *(v)    Gas transport in the firn / exchange with the atmosphere is already fully included in our model.*

*We agree that overall $^{14}C$ measurements in quartz are much better established than $^{14}C$ measurements in ice. However, we are not sure we agree that muogenic $^{14}C$ production rates in quartz are better characterized than muogenic $^{14}C$ production rates in ice in the published literature, particularly when $^{14}CO$ in ice is concerned. For quartz, there are the Heisinger et al (2002a, 2002b references in our manuscript) laboratory irradiation studies. However, the muon fluxes and energies used were not representative of fluxes / energy spectra in natural settings for rock and ice. There is also the Lupker et al 2015 study in a 15m deep rock core (also already cited in our manuscript) that had 9 samples in the depth zone (> ≈ 800 g cm$^{-2}$) where muogenic $^{14}C$ is expected to be dominant over neutron-produced $^{14}C$. These samples had an average relative uncertainty of 64% (based on comparing total $^{14}C$ content with overall uncertainty in*

*their Table 2). The Lupker et al analysis found that production rate via negative muon capture was in agreement with Heisinger et al (2002a) within uncertainties. For the fast muon mechanism production rate, Lupker et al. (2015) found a best-estimate value of zero, although their uncertainty range was large and included the Heisinger (2002b) value. We have also been informed (Greg Balco, open peer review process for Dyonisius et al., 2023) that there are further unpublished measurements in quartz that also support the Heisinger production rates.*

*In ice, the Dyonisius et al (2023) study measured 14CO, 14CO2 and 14CH4, had 10 depth levels where muogenic $^{14}C$ was dominant over neutron-produced $^{14}C$ and the 2-sigma relative uncertainty in results was 15%. The Hmiel et al study (this manuscript) examines $^{14}CO$ only, with 3 additional samples in ice and 6 in firn air lock-in zone where the muogenic signal is dominant, with much lower relative measurement uncertainties still. There are additional unpublished $^{14}CO$ data in ice and firn air from Law Dome, Antarctica from our collaborative group (currently in interpretation; Petrenko et al., 2023, AGU Fall Meeting) that are consistent with the Dyonisius et al and Hmiel et al muogenic $^{14}CO$ production rates. Finally, there was a study done by a different group (van der Kemp et al., 2002 reference in the manuscript) that measured $^{14}CO_2$ and $^{14}CO$ in ablating ice at Scharffenbergbotnen, Antarctica, using an entirely independent method (dry vs wet extraction, different facility and procedure for graphitization and $^{14}C$ measurement). Van der Kemp et al used much smaller sample sizes and their relative uncertainties are larger, but their results for muogenic production rates agreed well with those derived from Taylor Glacier (see Dyonisius et al., 2023). We would also note that the mass-depth range of ice studies is currently larger (to ≈10,000 g cm$^{-2}$ in this manuscript) than from published studies in quartz (≈4,000 g cm$^{-2}$) which is important for constraining the muogenic production rates.*

*To conclude, we would argue that measurements of $^{14}C$ in the gas phase in ice are well enough established at this point that the measurements themselves are not in doubt. While there is yet no clear explanation for the discrepancy between observed muogenic total $^{14}C$ production rates in quartz and in ice, we do offer a possibility in the new section (4.3.4) that this may have to do with unmeasured non-gas $^{14}C$ species in ice. We have clarified in both the abstract and conclusions that the findings of this study have to do with 14CO rather than total 14C, and also toned down the language regarding ice-quartz disagreement in every instance where it was discussed, also clarifying that the Dyonisius et al 2023 findings represented gas 14C species only (e.g., around line 110 in revised manuscript)*

Detailed:
L 54 ff. "The in situ produced 14C mainly forms 14CO2 and 14CO, with a smaller fraction forming 14CH4 and possibly other simple organics such as formaldehyde (Dyonisius et al., 2023; Hoffman, 2016; van der Kemp et al., 2002; Fang et al., 2021)." Of the total number of in-situ produced 14C atoms per gram ice, Hoffmann (2016) found a fraction of 11-25 % incorporated into the DOC fraction when performing a neutron irradiation experiment on Alpine ice core samples. The incorporation of cosmogenic in-situ 14C into DOC has later been supported by measurements in environmental samples (Fang et al., 2021). In Dyonisius et al., 2023 (and obviously all studies before 2022 or at least 2016) this partition has not been considered and is obviously also missing in the reviewed study here (see next point). Unlike 14CH4 this does not

seem to be a minor fraction and should be considered (maybe needed therefore: DOC concentrations, around 5 µg C / kg ice for polar ice; Preunkert et al. 2011). As you assume for CO, same partitioning for the n and muon mechanism needs to be similarly assumed here. Important might also be that the cosmogenic produced 14C incorporated into DOC is likely to behave differently in the firn/ice than the gaseous species (CO etc) in terms of diffusion and release into the porous firn (and transport therein), basically being fixated after incorporation (removed 14C partition in subsequent modeling of firn gas transport/retention/leakage).

*We clarified in the revised manuscript (around line 74) that $^{14}C$ partitioning into organics is is a substantial fraction. However, as discussed above, even at the maximum value of 25%, the organic fraction of in situ $^{14}C$ would be insufficient to explain the discrepancy in muogenic production rates between quartz and ice (an increase of ≈400% is needed for this). Since this study focuses on and measured $^{14}CO$ only, more detailed consideration of the organic $^{14}C$ fraction is beyond the scope of this work.*

L 319 ff. "We use a value of ΩCO= 0.31 for the fraction of total in situ 14C in ice that forms 14CO (Dyonisius et al., 2023; van der Kemp et al., 2002)." In Dyonisius et al. (2023), the value for ΩCO seems to be 33.7 % associated with an uncertainty of ±11.4 %. Has an uncertainty for ΩCO been considered and propagated here? This seems relevant considering the narrow range of µ- and µf and the difference compared to earlier values (see Table 2). If not, this should not be too difficult to be introduced for example in a similar way as done for the uncertainty of the n production rate (introduction of an additional, adjustable "uncertainty factor" like Fn in L 326). Also see above regarding the missing partition which is incorporated into DOC.

*The $f_{µ-}$ and $f_{µf}$ dimensionless factors account for both the fraction of $^{14}C$ that forms $^{14}CO$ and the reduction in muogenic production rates from the Heisinger values. This was already stated in the original manuscript on lines 364 – 365. The f-value uncertainty ranges thus already incorporate any uncertainty in $Ω^{CO}$. The new table (Table 1 in the revised manuscript) further clarifies this. As mentioned above, incorporating the organic 14C fraction into the model is beyond the scope of this work.*

L 356,357 "...(we use α=0.75, consistent with Dyonisius et al., 2023 and Heisinger et al., 2002b),...". Heisinger et al. (2002b) considered uncertainties in the muon energy and flux to be in the range of 10% each. It does not seem you similarly considered and propagated these uncertainties. Aiming for a comprehensive study, propagating all uncertainties will finally be more useful, resulting in the most realistic range estimates. Especially if there still persist a number of unknowns (i.e. a detailed understanding of all process and mechanisms involved). If not considered, the factors or uncertainties should at least be summarized in the discussion (reference to suggested additional table?), providing a possible tie-point for future studies.

*The referee is correct that we did not consider uncertainties in the muon fluxes and mean energies. However, as the referee mentions, these uncertainties are expected to be relatively small as there is a good amount of observations available from muon detectors underground, underwater and under ice. As we use the same parameterizations of these fluxes / energies as studies in quartz (model from Balco et al., 2008), this could not explain the ice – quartz discrepancy in muogenic $^{14}C$ production rates. The uncertainties that were / were not considered are now clearly summarized in the new Table 1. In the conclusion in the revised manuscript, we*

*mentioned exploring these uncertainties as a point of improvement for future studies, as the referee suggests.*

L 378 ff. "We introduce these two reservoirs because a preliminary analysis showed that using a single ice grain reservoir does not provide a good fit to the observations." Is there any hypothesis of a possible mechanism/process to justify this partitioning into two reservoirs? What is considered as a good fit (especially if considering all other uncertainties, including the ones mentioned above and some of the points mentioned below), i.e. how bad would the fit be?
*Please see our detailed response to referee 1 on this point.*

Section 4.2, L 481 ff. Is ventilation not part of gas transport/movement in the firn? Since you are using a frin transport model, I do not see why your proposed mechanism would not also (to some extent at the least) include ventilation. Further, snow metamorphism (the most common type of recrystallization of snow and the uppermost firn) is a very fast process (days to weeks); e.g. Pinzer et al., 2012. With depth and higher density of the firn, the recrystallization process will become slower, probably in the order of a few years at the LIZ (e.g. Duval et al., 1995). Thus, would it not be more appropriate to, in the model, combine the processes of diffusion and metamorphism / recrystallization, essentially leading to an "enhanced diffusion" (strong enhancement in the upper firn and much less close/at the LIZ)? Maybe a partitioning of the model into two reservoirs could then be avoided?
*We thank the referee for pointing this out – this was not phrased as clearly as it could have been in the manuscript. Prior studies discussed processes such as recrystallization and sublimation **in combination with** wind ventilation to explain $^{14}$C loss from ice grains in the firn. We clarified this in the revised manuscript (now on line 548). Gas diffusion by itself (even without recrystallization) predicts such rapid loss of $^{14}$CO that $^{14}$CO content of ice grains in the firn would be expected to be essentially zero all the way through the firn column. So the second reservoir is needed to ensure some $^{14}$CO retention, consistent with the measurements. While we agree that a model that is based fully on physical processes would be better, so far we just have a few data points from a single site to guide the interpretation of this process. More data, from multiple sites in future studies would be needed to confirm the results and better identify the $^{14}$C loss / retention processes. We added a sentence in the conclusions in the revised manuscript to highlight this.*

**Minor issues**
L 187, equation 1. Definition of xCO seems to be missing in the text.
*This was already defined on lines 104 – 105 in the original manuscript*

L 188 "...pMC is the sample or blank 14C activity in pMC units...". pMC denotes percent modern carbon, a pMC unit does not exist (as the name tells, it is a percentage).
*We made this slight edit in the revised manuscript*

L 191 (& equation 1) "1.1694 × 10-12 is the 14C / (13C + 12C) ratio corresponding to the absolute international 14C standard activity (Hippe and Lifton, 2014), ..." pMC is defined based

on a half life of 5730 (as you mention elsewhere, the half life you used), but Hippe and Lifton, in their reformulation, from which your value of 1.1694 × 10-12 results, considered a half-life of 5700. Note that their reformulation is performed for two main reasons: (i) to omit a necessary correction in activity for the decay from 1950 to the year of measurement of the international 14C standard used for AMS calibration (yielding the number for the activity to be used with 1.1694 × 10-12) and (ii) to account for the conventionally introduced 13C normalization for AMS measurements (to -25 per mil; not to be confused with the additionally performed normalization of the standard to -19 per mil to account for the switch to a new material of the international AMS reference standard; see e.g. https://www.hic.ch.ntu.edu.tw/AMS/A%20guide%20to%20radiocarbon%20units%20and%20calculat ions.pdf). In any case, the reformulation eventually leads to their Eq 22, which in principle corresponds to your Eq 1 (without the terms accounting for CO molecules per volume). Your equation misses one important term (since this is one of the two main reasons to perform the reformulation in the first place), Ab12s, which results from accounting/correcting for 13C normalization. I suggest checking carefully (maybe using the activity of the reference standard and correcting to the year of measurement might be more appropriate for your data anyhow; not that all that matters much with regards to all other uncertainties, I think...).

*We have re-checked our Equation 1 and think that it is the correct form to use for our samples. The full derivation of this equation is available in the Supplement to the Petrenko et al., 2016 reference in the manuscript. pMC is commonly used for reporting $^{14}C$ results, and is how ANSTO reports theirs. A decay correction from 1950 is needed to get from pMC to absolute $^{14}C$ abundance (exponential term in our Equation 1), and using a half-life of 5700 vs 5730 years in this decay correction changes the result by less than 0.01%.*

L 297 "Figure 1 illustrates total 14C production rates by each mechanism versus depth at Summit, ...". The term "each mechanism" might not be ideal with regards to the previous sentence where also the mechanism of "loss of this 14C from the ice grains via leakage into the open porosity (firn air) or closed porosity (air bubbles)." is mentioned. Maybe clearer would be, "Figure 1 illustrates total 14C production rates by secondary cosmic ray neutrons and muons versus depth at Summit, ...", even though not the most pretty due to repetition.
*We have made this edit in the revised manuscript (now on line 327).*

L 355 "...β(h) is a unitless depth dependence factor...". Shouldn't that be mass-depth dependence factor?
*We made this edit in the revised manuscript (now on line 393)*

**References**
Lupker, M and 7 others (2019) In-situ cosmogenic 14C analysis at ETH Zürich: characterization and performance of a new extraction system. Nuclear Instruments and Methods in Physics Research Section B: Beam Interactions with Materials and Atoms 457(July), 30–36. doi: 10.1016/j.nimb.2019.07.028
Nichols KA. A decade of in situ cosmogenic 14C in Antarctica. Annals of Glaciology. 2022;63(87-89):67-72. doi:10.1017/aog.2023.13

Susanne Preunkert, Michel Legrand, P. Stricker, S. Bulat, Irina Alekhina, et al.. Quantification of Dissolved Organic Carbon at Very Low Levels in Natural Ice Samples by a UV-Induced Oxidation Method. Environmental Science and Technology, 2011, 45 (2), pp.673-678. 10.1021/es1023256.insu- 00605044

Pinzer, B. R., Schneebeli, M., and Kaempfer, T. U.: Vapor flux and recrystallization during dry snow metamorphism under a steady temperature gradient as observed by time-lapse micro-tomography, The Cryosphere, 6, 1141–1155, https://doi.org/10.5194/tc-6-1141-2012, 2012.

P. Duval, O. Castelnau. Dynamic Recrystallization of Ice in Polar Ice Sheets. J. Phys. IV,1995, 05 (C3), pp.C3-197-C3-205. 10.1051/jp4:1995317. jpa-00253683

Whang et al., J. Phys. Chem. A, 2023, 127, 2, 517–526, https://doi.org/10.1021/acs.jpca.2c07554

Y. S. Yi, Y. Han, K. D. Kwon, S. K. Lee and S. D. Hur ACS Earth and Space Chemistry 2021 Vol. 5 Issue 11 Pages 3258-3267 DOI: 10.1021/acsearthspacechem.1c00308 https://doi.org/10.1021/acsearthspacechem.1c00308

***Other changes not described above:***
- *Some minor corrections to formatting / typos*
- *Added the Petrenko et al 2017 reference that was omitted by mistake in the original manuscript*
- *Replaced BenZvi et al 2019 reference (which was a conference proceedings, not peer reviewed) with Petrenko et al 2024 which was just accepted in The Cryosphere*
- *The position of some figures end tables was adjusted as needed to accommodate added text / table*
- *Present addresses for some authors were updated, resulting in renumbering of some affiliations*

**5 June 2024**
**Additional responses regarding technical corrections requested by the editor:**
Dear authors

Thank you for your careful consideration of the reviewer comments. Reading the revised version of your manuscript, I got convinced that in its revised form the manuscript does not have to go through a further round of reviews. I have a few technical corrections that I would ask you to include before publication in addition to the changes you are already suggested.

1. On page 8 line 308 of your tracked-change version of the manuscript, you say that the firn models do usually not include the thinning by glacier flow. I would recommend to explicitly say at this point that it does include thinning by densification but no additional thinning of the layer thickness in WE depth by deformation. Just not to confuse the outsider.
*This has been clarified in the further revised manuscript*

2. I think the issue about the used standard activity (1.1694 x 10^-12) and the half life used (5700 vs 5730 yr) is not cleared up sufficiently yet.

For example, for the decay constant you use 1.210 x 10-4 1/yr, while in Petrenko et al , 2014 1.216 < 10^-4 1/yr is used. The latter is also the value by Hippe and Lifton referring to a 14C half life of 5700 years. So your formula seems not to be consistent with Petrenko et al 2014. Please clear up this inconsistency. But I agree it does not change the conclusions.

*We thank the editor for catching this, this is indeed an inconsistency that we missed in responding to the referee about Equation 1. Changing the decay constant to $1.216 \times 10^{-4}$ $yr^{-1}$ in Equation 1, to be consistent with both the $1.1694 \times 10^{-12}$ value we use (for the $^{14}C$ / ($^{13}C$ + $^{12}C$) ratio corresponding to the absolute international $^{14}C$ standard activity) and with Petrenko et al. (2016) changes the sample 14CO values by 0.004%, without an impact on the values shown in the figures and tables. For consistency, we changed the decay constant value to $1.216 \times 10^{-4}$ $yr^{-1}$ in our description of Equation 1 in the revised manuscript.*

3. page 18 line 595: It is not obvious why diffusion of CO from microbubbles into the open firn porosity would be slower than from CO in the ice grain. This needs an explanatory sentence.

*Further movement of $^{14}CO$ from microbubbles to open porosity would be controlled by the gas permeation coefficient (product of diffusivity and solubility). As solubility of gases in ice is very low (e.g., Patterson and Saltzman, 2021 and references therein), this process is slow. This is different from the situation of $^{14}CO$ immediately following in situ production, as in this case $^{14}CO$ is already in the ice lattice and only diffusivity in ice matters. We have added this explanation in the further revised manuscript.*

page 19 line 616: "In each case, the reduced..."
*We have made this slight edit in the further revised manuscript*